# Effects of Dietary Supplementation with High Fiber (Stevia Residue) on the Fecal Flora of Pregnant Sows

**DOI:** 10.3390/ani10122247

**Published:** 2020-11-30

**Authors:** Miao Yu, Tie Gao, Zhen Liu, Xinping Diao

**Affiliations:** College of Animal Science and Technology, Northeast Agricultural University, Harbin 150030, China; YM161129@163.com (M.Y.); gt13204601339@163.com (T.G.); qq1684285293@163.com (Z.L.)

**Keywords:** high-fiber diets, stevia residue, pregnant sows, fecal flora

## Abstract

**Simple Summary:**

This study aimed to investigate the effects of different supplementation levels of stevia residues in high-fiber diets on the fecal microorganisms of pregnant sows. Stevia residue supplementation significantly increased the abundance of beneficial bacteria, such as *g__Lachnospiraceae_XPB1014_group*, *g__Christensenellaceae_R-7_group*, and *g__Ruminococcaceae_UCG-005* (*p* < 0.05), and significantly reduced the abundance of harmful bacteria, such as *Treponema_2* (*p* < 0.05), at the genus level. The stevia-residue supplementation reduced the feed cost, with the highest reduction rate being 13.33%. In conclusion, high-fiber diets can improve the intestinal balance and evenness index of the intestinal flora in pregnant sows, promote the relative abundance of beneficial bacteria, reduce the relative abundance of harmful bacteria, and reduce feed cost.

**Abstract:**

This study aimed to investigate the effects of different supplementation levels of stevia residues in high-fiber diets on the fecal microorganisms of pregnant sows. Forty-eight first-farrowing Danish Landrace sows with similar weight, age, and due date were randomly divided into four groups. The control group was fed a normal diet (CON; 3.15% crude fiber content), and the treatment groups were supplemented with 20% stevia residue (SRL), 30% stevia residue (SRM), or 40% stevia residue (SRH) stevia residue, with crude fiber content of 7.79%, 9.15%, and 10.68%, respectively. The test period was 135 days, and Illumina Miseq high-throughput sequencing was used to test the fecal bacteria of the pregnant sows on day 56. Compared to the control group, species diversity was greater in the 30% stevia residue group. At the phylum level, Firmicutes and Spirochaetes had the greatest relative abundances; Firmicutes was most abundant in the SRM group, and Spirochaetes was most abundant in the CON group. At the genus level, *Lactobacillus*, *Streptococcus*, and *Lachnospiraceae_XPB1014_group*, and *Treponema_2* had the highest relative abundances in the stevia-residue treatments. Among them, *Lactobacillus* and *Treponema_2* were more abundant in SRL, *Streptococcus* was more abundant in SRH, and *Lachnospiraceae_XPB1014_group* was more abundant in SRM. Linear discriminant analysis effect size (LEfSe) showed that the bacterial species differed among the stevia residue treatments. SRL mainly included *g__Lactobacillus* and *g__Romboutsi*, SRM mainly included *g__Lachnospiraceae_XPB1014_group* and *g__Ruminococcaceae_UCG_014*, SRH mainly included *g__Streptococcus*, and CON mainly included *g__Treponema_2*, *f__Clostridiaceae_1*, *g__norank_f__Muribaculaceae*, and *g__norank_f__p_251_o5*. Stevia-residue supplementation significantly increased the abundance of beneficial bacteria, such as *g__Lachnospiraceae_XPB1014_group*, *g__Christensenellaceae_R-7_group*, and *g__Ruminococcaceae_UCG-005* (*p* < 0.05), and significantly reduced the abundance of harmful bacteria, such as *Treponema_2* (*p* < 0.05). Moreover, compared with the control group, the stevia-residues treatment groups reduced the feed cost 8.33%, 12.50%, and 13.33% per sow per day, respectively. In conclusion, high-fiber diets can improve the intestinal balance and evenness index of the intestinal flora in pregnant sows, promote the relative abundance of beneficial bacteria, reduce the relative abundance of harmful bacteria, and reduce feed cost. The optimal supplementation level of the stevia residue was found to be 30%.

## 1. Introduction

Pig farms began to restrict pregnant sow feeding in the early 1980s, which reduced embryonic mortality, prevented excessive fat deposition in the breast tissue (causing postpartum dairy-free and fatty milk), and promoted breast cell proliferation [1]. Restricted pregnant sow feeding aids in production, but it is also associated with negative feeding effects. Constipation frequently occurs, and in severe cases, results in prolapse of the anus, systemic diseases, and stereotyped behavior, such as empty chewing, rail biting, and ground licking [2]. High-fiber diets are used to balance the nutritional and feed intake levels, to improve the reproductive performance of sows. Stevia residue is the waste of stevia after stevioside extraction, dehydration, and high-temperature drying. Most stevia residue is landfilled and burned, and a small part is made into fertilizer, which causes great waste and environment pollution [3]. Stevia residue has a low price, high fiber content, high crude-protein content, and long shelf-life; it is a source of high-quality green crude fiber, and crude fiber can accelerate intestinal peristalsis and emptying [4], reduce feed nutrients, and effectively solve the problems of pregnant sows breeding. However, the effects of high-fiber diets on the intestinal microflora of pregnant sows have rarely been investigated. Studies suggest that the fermentation of soluble fiber can increase microbial growth, while insoluble fiber promotes chyme evacuation and reduces the available time for microbial growth [5]. To examine the effects of high-fiber diets on the intestinal flora of pregnant sows, this study fed pregnant Danish Landrace sows similar caloric–protein ratios with different crude fiber contents and compared the effects of crude fiber on the fecal microflora.

## 2. Materials and Methods

The animal care and treatment procedures used in this study complied with the standards for the care and use of laboratory animals of Northeast Agricultural University (2011–2019). The animals were maintained, according to the National Research Council Guide (1996), in metabolic cages, and had free access to food and water. This study was performed in strict accordance with the recommendations of the National Research Council Guide (1996), and the Ethical and Animal Welfare Committee of Heilongjiang Province, China, approved all of the animal experimental procedures.

### 2.1. Experimental Materials

The experimental pigs were provided by Zuoyou Animal Husbandry (Hegang, Heilongjiang Province, China). The experimental diets were provided by Heilongjiang Anyou Biotechnology Co., Ltd. (Qiqihar, Heilongjiang Province, China). Stevia residue was provided by the Harbin Huada Feed Company (Harbin, Heilongjiang Province, China). Stevia residue is a gray-black powder with a characteristic chrysanthemum odor that is mixed with some straw-like substance (Figure 1).

### 2.2. Animals and Experimental Design

Forty-eight first-farrowing Danish Landrace sows of similar weight, age, and due date were randomly divided into 4 groups, with 12 repetitions per group and 1 sow per repetition (Table 1). The experimental period lasted 135 days from breeding to weaning of the piglets. The sows were fed prepared corn–soybean-meal diets with different crude fiber levels, according to the Chinese ministry of agriculture of pig-breeding standards (NY/T 65-2004) and the feeding levels of Anyou Biological Technology Co., Ltd. (Heilongjiang Province, China) and Zuoyou Animal Husbandry (Heilongjiang Province, China). The stevia-residue supplementation levels are shown in Table 2, and the diet composition and nutrition levels are shown in Table 3. The control group (CON) was fed a basal diet, and the experimental groups were fed 20% stevia residue (SRL), 30% stevia residue (SRM), or 40% stevia residue (SRH). The crude fiber contents were 3.15% (CON), 7.79% (SRL), 9.15% (SRM), and 10.68% (SRH). Moreover, protein/energy ratios were similar between diets.

The pig house was cleaned and disinfected before the experiment, and during the experiment, the pig house was cleaned every day, at 07:00 and 14:00, with povidone iodine Biocid-30, Pfizer Animal Health Products Co., Ltd., Shanghai, China. Seven days before the expected delivery date, the sows were transferred to the delivery house, with a raised bed. The same breeder was responsible for feeding and management throughout the experimental period. The sows’ feed was restricted at different levels, from estrus to 110 days of pregnancy. Each sow was fed twice daily, at 06:00 and 15:00, and had free access to a duck-bill water device. During the whole experimental period, immunizations were administered according to the scheme of the pig farm, and changes in the diet were based on the farm’s normal feeding regime. The control group and the experimental groups were feed restricted according to the pig farm’s management. Control sows were fed 2.2 ± 0.2 kg/d from breeding until 35 days of pregnancy and 3.0 ± 0.2 kg/d from 36 days of pregnancy until delivery. The experimental groups were fed 3.0 ± 0.2 kg/d from breeding until 35 days of pregnancy and 4.0 ± 0.2 kg/d from 36 days of pregnancy until delivery. The control and experimental groups were controlled to same metabolizable energy (ME), by different feed intake, and adjusted according to the body size and back-fat thickness, and the remaining feed was collected at 22:00 every day.

The statistics of feed cost is based on the fact that there is no significant difference in reproductive performance. The table of reproductive performance is shown in the Appendix A.

### 2.3. Sample Collection

On day 56 of pregnancy, fresh fecal samples were collected from 6 randomly selected sows from each group. The samples were placed in 5 mL sterile tubes and stored at −20 °C, until analyses of the microbial levels. 

### 2.4. High-Throughput Sequencing of the Intestinal Microorganisms

The frozen fecal samples were sent to Shanghai Meiji Biomedical Co., Ltd. (Shanghai, China), for high-throughput sequencing. The main experimental materials and reagents are shown in Table 4, and the detailed experimental instruments and equipment are shown in Table 5. The E.Z.N.A.^®^ Soil DNA Kit (Omega Bio-Tek, Norcross, GA, USA) was used to extract total microbial community genomic DNA, according to the manufacturer’s instructions. The DNA extract was checked on a 1% agarose gel, and the DNA concentration and purity were determined on a NanoDrop 2000 UV–vis spectrophotometer (Thermo Scientific, Wilmington, DE, USA). The DNA extract was checked by gel electrophoresis, on a 1% agarose gel, at 5 V/cm for 20 min. The hypervariable V3–V4 region of the bacterial 16S rRNA gene was amplified with primers 338F (5′-ACTCCTACGGGAGGCAGCAG-3′) and 806R (5′-GGACTACHVGGGTWTCTAAT-3′), in an ABI GeneAmp^®^ 9700 PCR thermocycler (ABI, CA, USA). The PCR mixture contained 4 μL of 5× TransStart FastPfu buffer, 2 μL of 2.5 mM dNTPs, 0.8 μL each of the forward and reverse primers (5 μM), 0.4 μL of TransStart FastPfu DNA Polymerase, 0.2 μL of bovine serum albumin (BSA), 10 ng of the template DNA, and ddH_2_O up to a total reaction volume of 20 μL. The PCR reactions were performed in triplicate, with the following cycling conditions: initial denaturation at 95 °C for 3 min, followed by 27 cycles of denaturing at 95 °C for 30 s, annealing at 55 °C for 30 s, and extension at 72 °C for 45 s, and a single final extension at 72 °C for 10 min. The PCR product was extracted from a 2% agarose gel and purified by using the AxyPrep DNA Gel Extraction Kit (Axygen Biosciences, Union City, CA, USA) and quantified on a Quantus™ Fluorometer (Promega, Madison, WI, USA). The Illumina Miseq PE300 platform was used for sequencing (Shanghai Maggi Bio-Pharmaceutical Technology Co., Ltd., Shanghai, China).

### 2.5. Statistical Analysis

The raw 16S rRNA gene sequencing reads were demultiplexed, quality-filtered in fastp version 0.20.0 [6], and merged in FLASH version 1.2.7 [7]. The operational taxonomic units (OTUs) with a 97% similarity cutoff [8,9] were clustered with UPARSE version 7.1 [8], and the chimeric sequences were identified and removed. The taxonomy of each OTU representative sequence was analyzed by RDP Classifier version 2.2 [10] against the 16S rRNA database (e.g., Silva v138), using a confidence threshold of 0.7. The threshold for statistical significance was *p* < 0.05.

## 3. Results

### 3.1. Sequencing Data Estimation

A total of 24 samples were collected, including 973,036 effective sequences, 405,905,731 effective bases, and an average length of 417.1539 bp. The sequence length ranged from 401 to 440 bp, of which 47.28% were 401 to 420 bp, and 52.72% were 421 to 440 bp. The number of species in the samples was determined by OTU ranking at the taxonomic level, on the abscissa axis, and the relative percentage of species at the taxonomic level, on the ordinate axis, which generated an OTU rank–abundance curve (Figure 2). There were no curve differences in the CON, SRL, SRM, or SRH groups. With increasing rank values, the abundances tended to gradually level off, indicating species evenness and better richness. CON and SRL had similar species diversity, SRM had higher species diversity but a less dominant population, and SRH had lower species diversity and a more dominant population.

### 3.2. Alpha Diversity Analysis

Alpha diversity analysis can indicate the richness and diversity of species communities. Sobs, Chao, and Ace indices are used to assess community richness, and the Sobs index is an indicator of the actual richness. The Shannon and Simpson indices reflect community diversity—a higher Shannon index and lower Simpson index indicate greater microbial diversity. The Coverage index reflects community coverage; a greater Coverage index infers a higher probability that the sequence in the samples will be measured. As seen in Table 6, the Coverage index of each group was greater than 0.99, indicating that the probability of undetected sequences in the samples was very low. SRH had the highest Sobs index (651.50), and SRL had the lowest (603.83). SRH and SRL significantly differed (*p* < 0.05), but the other groups did not (*p* > 0.05). There were no differences in the Chao or Ace indices (*p* > 0.05). The Shannon index was calculated separately, and the community diversity did not differ among groups (*p* > 0.05), but SRL, SRM, and SRH tended to be more evenly distributed. The Simpson index for SRH and SRM was significantly higher than for SRL (*p* < 0.05). SRL had the highest community diversity (Shannon index = 4.00, Simpson index = 0.05), while SRH had the lowest community diversity (Shannon index = 3.92, Simpson index = 0.08).

Using the measured data from the Sobs and Shannon indices, we used sequences of known OTU ratios and the corresponding OTU number expectations to construct dilution curves (with random sequencing data as the abscissa and the number of species (Sobs index) and diversity index (Shannon index) as the ordinates). Sobs index and Shannon index dilution curves can be used to evaluate if the sequenced amount is enough to cover all taxa and indirectly estimate the diversity of species in the samples. The Sobs index dilution curve is shown in Figure 3, and the Shannon index dilution curve is shown in Figure 4. With increased effective sequencing depth, the Sobs and Shannon index dilution curves rapidly increased and then flattened, suggesting an adequate amount of sequencing data, good sequencing quality, and appropriate depth and representativeness. The sequencing depth was enough to cover all species in the samples. There were no differences in the number of species in CON, SRL, and SRH (*p* > 0.05), while the number of species in SRM was significantly reduced (*p* < 0.05). The diversity of species did not differ between groups (*p* > 0.05).

### 3.3. Fecal Flora Composition Analyses

#### 3.3.1. Venn Diagram Analysis of the Fecal Flora

Venn diagrams can be used to examine the unique and common OTUs in each group, and the similarity and overlap of OTUs in different environmental samples can be intuitively presented. As shown in Figure 5, the number of OTUs in CON, SRL, SRM, and SRH was 856, 918, 893, and 921, respectively, and there were 667 common OTUs in the four groups. The number of unique OTUs in CON, SRL, SRM, and SRH were 32, 5, 10, and 13, respectively. SRL shared 768 OTUs with CON and had 150 unique OTUs. SRM and CON shared 740 OTUs, and SRM had 153 unique OTUs. Finally, SRH and CON shared 748 OTUs, and SRH had 173 unique OTUs.

#### 3.3.2. Composition Analysis of the Fecal Flora

##### Relative Distribution of the Fecal Microorganisms at the Phylum Level

Based on the community abundances of the OTU samples, differences in the samples at the phylum level were analyzed, and the results are presented in Figure 6 and Figure 7. The bacteria in the 24 samples mainly included Firmicutes, Spirochaetes, Bacteroidetes, Actinobacteria, and Verrucomicrobia. The fecal flora composition of the pregnant sows was dominated by Firmicutes, while the relative abundance of Firmicutes was SRM > SRH > SRL > CON, accounting for 73.04%, 82.21%, 84.49%, and 83.12%, respectively. The relative abundance of Spirochaetes was CON > SRL > SRH > SRM. Compared to CON, the abundances of Spirochaetes in SRL, SRM, and SRH decreased by 23.22%, 77.69%, and 50.65%, respectively. The relative abundance of Bacteroidetes was CON > SRL > SRH > SRM, decreasing by 64.76%, 84.90%, and 80.80%, respectively, compared to CON. The relative abundance of Actinobacteria was SRM > SRL > SRH > CON, accounting for 0.40%, 1.71%, 1.45%, and 1.89% in each group, respectively. Verrucomicrobia had a very low abundance in CON, SRL, and SRH, and the relative abundance of Verrucomicrobia was SRM > SRL > SRH > CON. Verrucomicrobia was most abundant in SRM and increased significantly (1.31%) compared to the other three groups. Bacteria accounting for less than 0.01% of the total were classified as others. These results indicate that the dominant bacteria in pregnant sows are Firmicutes and Spirochaetes, and that Firmicutes was most abundant in the SRM group (30% stevia residue). Spirochaetes was most abundant in the CON group.

##### Relative Distribution of the Fecal Microorganisms at the Genus Level

Based on the community abundances of the OTU samples, differences in the samples at the genus level were analyzed, and the results are presented in Figure 8 and Figure 9. The fecal flora of the pregnant sows are mainly composed of *Lactobacillus, Streptococcus*, *Lachnospiraceae_XPB1014_group*, and *Treponema_2*, accounting for 54.56%, 55.98%, 54.76%, and 56.2% of the total in each group, respectively. The relative abundance of *Lactobacillus* was SRL > SRH > CON > SRM, accounting for 28.83%, 30.62%, 17.84%, and 28.84%, respectively. The relative abundance of *Streptococcus* was SRH > SRM > SRL > CON, accounting for 5.34%, 9.47%, 13.70%, and 17.84%, respectively. The relative abundance of *Lachnospiraceae_XPB1014_group* was SRM > CON > SRL > SRH, accounting for 7.61%, 5.90%, 20.33%, and 3.05%, respectively. The relative abundance of *Treponema_2* was CON > SRL > SRH > SRM, accounting for 12.78%, 9.99%, 2.89%, and 6.42%, respectively. Species with relative abundances more than 0.5% were *Romboutsia, Christensenellaceae_R-7_group, Clostridium_sensu_stricto_1, Ruminococcaceae_UCG-005, Ruminococcaceae_UCG-014, unclassified_f__Lachnospiraceae, norank_f__Muribaculaceae, norank_f__p-251-o5, unclassified_o__Lactobacillales, Turicibacter, Ruminococcaceae_NK4A214_group*, *Terrisporobacter, Rikenellaceae_RC9_gut_group*, *Marvinbryantia, Coprococcus_3*, and *Akkermansia*. Bacteria that accounted for less than 0.01% of the total were classified as others. These results indicate that *Lactobacillus* and *Treponema_2* were more abundant in SRL, *Streptococcus* was more abundant in SRH, and *Lachnospiraceae_XPB1014_group* was more abundant in SRM.

### 3.4. Linear Discriminant Analysis Effect Size (LEfSe)

When the linear discriminant analysis (LDA) score is greater than 4, there is a statistical difference between groups for a given species. LEfSe analysis was used to examine the feces of sows fed different amounts of stevia residue. As shown in Figure 10 and Figure 11, the bacteria varied according to the amount of stevia residue in the diet. The SRL group mainly included *g__Lactobacillus* and *g__Romboutsia*, SRM mainly consisted of *g__Lachnospiraceae_XPB1014_group* and *g__Ruminococcaceae_UCG_014*, SRH mainly included *g__Streptococcus*, and CON mainly consisted of *g__Treponema_*2, *f__Clostridiaceae_1*, *g__norank_f__Muribaculaceae*, and *g__norank_f__p_251_o5*. The genera significantly differed between groups (*p*-values of genera with significant differences were all less than 0.05; Table 7).

## 4. Discussion

In recent years, the intestinal health of pigs has received greater attention. The small intestines are the site of feed digestion and nutrient absorption; undigested and indigestible substances in the small intestines are decomposed by microorganisms of large intestines. The intestines represent the largest immune organ in pigs, with 70% of the immune cells and 60–80% of the immunoglobulin in the body. The intestinal tract is one of the largest habitats for microbes, and some have even called it a microbial organ [11]. In addition to the beneficial microorganisms, harmful bacteria are also widely abundant in the intestines, accounting for 70% of the diseases in animals [12,13,14]. Under normal conditions, the intestinal flora maintains the balance between the internal and external environments, ensures normal digestion and nutrient absorption, resists foreign pathogen invasion, and produces important metabolites and bioactive components [15]. High-fiber diets had higher crude fiber contents, including more dietary fiber than conventional diets in pig farms. Dietary fiber is prebiotic and changes the intestinal microbiota and promotes the growth of *Bifidobacteria* and *Lactobacillus* [16]. Li et al. [17] found adding 1% Gracilaria Residue to the diet of pregnant sows can increase the number of *Lactobacillus* and reduce the number of *Escherichia coli* in rectum. Liu [18] reported feeding pregnant Meishan sows in three different crude fiber content diets (2.5% CF group, 5% CF group, and 7.5% CF group), compared with 2.5% CF group, the operational taxonomic units (OTUs) was higher than that of the 5% CF group and 7.5% CF group obviously (*p* < 0.05), the fecal bacteria community diversity improved significantly (*p* = 0.01), and the relative abundances of *Bacillibacteria*, *Bacteroides*, *Fibrobacteres*, *Ruminococcus*, *Butyrivibrio*, and *Lactobacillus* of 7.5% CF group were significantly increased compared to those of the 2.5% CF group (*p* < 0.05). The relative abundances of *Streptococcus* and *Escherichia-Shigella* of 7.5% CF group were significant decreased than those of 2.5% CF group (*p* < 0.01, *p* = 0.04). Experiments have shown that higher fiber content can increase the abundance of Firmicutes in pregnant sows. Xu et al. [19] found that adding 2.0% guar gum plus pregelatinized waxy maize starch (SF) in pregnant sows diets can significant increased gut bacteria community diversity, and *Firmicutes* and *Ruminococcaceae* were obviously enriched in SF-fed sows (*p* < 0.01). Zhou et al. [20] observed that, compared with the two control groups, the 1.5% inulin groups had an extremely significant increase in the fecal microbial community diversity (*p* < 0.01). Moreover, the dominated phyla were significant increased, including *Firmicutes*, *Bacteroidetes*, *Spirochaetes*, *Tenericutes*, and *Proteobacteria* (*p* < 0.05), and had the tendency toward the increase of the relative abundance of *Firmicutes*/*Bacteroidetes* ratio (*p* = 0.07). However, Guo [21] found that high-fiber treatments in Tianjing black pigs can significantly improve Firmicutes abundance in the jejunum and ileum (*p* < 0.05) and that Firmicutes will first increase and then significantly decrease with increasing fiber levels in the cecum and duodenum (*p* < 0.05). It might have the same tendency for gestation sows, as the reduced abundances may have been due to the exorbitant coarse fiber contents (15.5%); the microbes in large intestines cannot digest such high concentrations of crude fiber, which may have caused microecological imbalances. The results from the above experiments are consistent with this study. *Lactobacillus* and *Bacillus* (Firmicutes) play important roles in maintaining intestinal health and promoting intestinal physiological function. Increased Firmicutes abundances suggest that high-fiber diets can maintain intestinal microecological balance. High dietary fiber levels will lead to an increase in the number of fiber-breaking bacteria) [22]. Increased numbers of fiber-digesting bacteria can increase the utilization of dietary fiber in the large intestine of gestating sows. Wang et al. [23] showed that castrated boars fed a 15% alfalfa fiber diet increased the abundances of *Lactobacillus* and *Desulfovibrio*. Laitat et al. [24] also reported that dietary supplementation with 24% beet residue could significantly increase the relative abundance of *Bifidobacteria* and *Lactobacillus* in the intestinal tract of growing pigs, while reducing *Escherichia coli* abundance. This is consistent with the conclusions of our study. Liu et al. [25] found that *Christensenellaceae_R-7_group* abundance was higher on days 14 and 28 in weaned piglets fed brown algae glume, and *Christensenellaceae_R-7_group* abundance was significantly higher on day 14 (*p* < 0.05). This is consistent with the significant increases in *g__Christensenellaceae_R-7_group* abundance in this experiment. *Christensenellaceae_R-7_group* can regulate lipid metabolism and reduce the occurrence of obesity [26], so high-fiber diets may help to prevent obesity in pregnant sows.

## 5. Conclusions

A high-fiber diet can maintain intestinal homeostasis in pregnant sows, promote the relative abundance of beneficial bacteria, and reduce the relative abundance of harmful bacteria. The 30% stevia residue group with a crude fiber content of 9.15% had the best effects. Stevia residue is low-cost and high yielding, which can effectively reduce the feed cost of pregnant sows.

## Figures and Tables

**Figure 1 animals-10-02247-f001:**
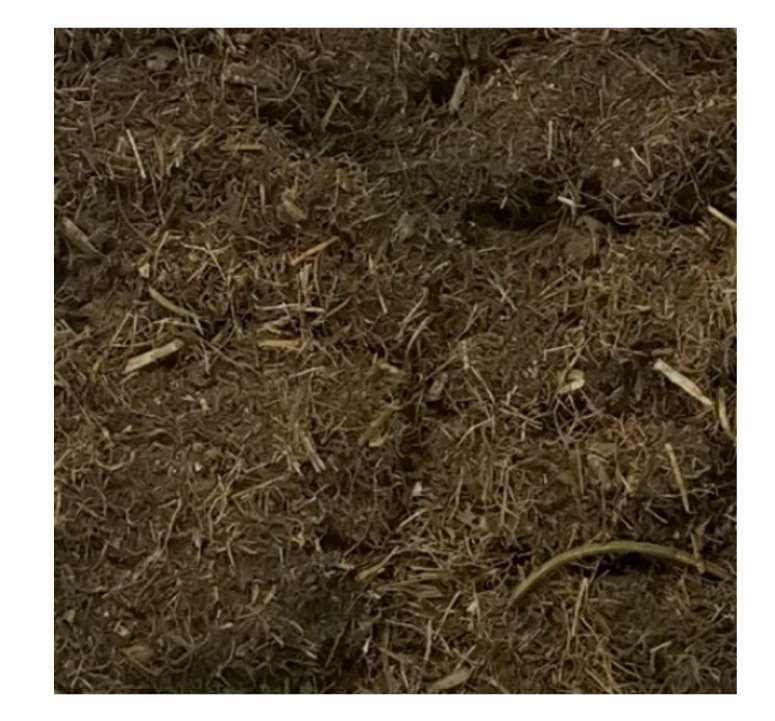
Stevia residue.

**Figure 2 animals-10-02247-f002:**
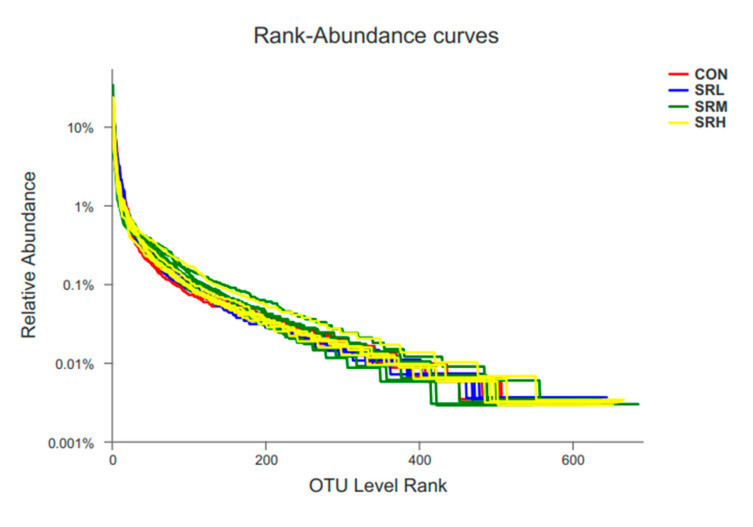
Operational taxonomic units (out) rank–abundance curve.

**Figure 3 animals-10-02247-f003:**
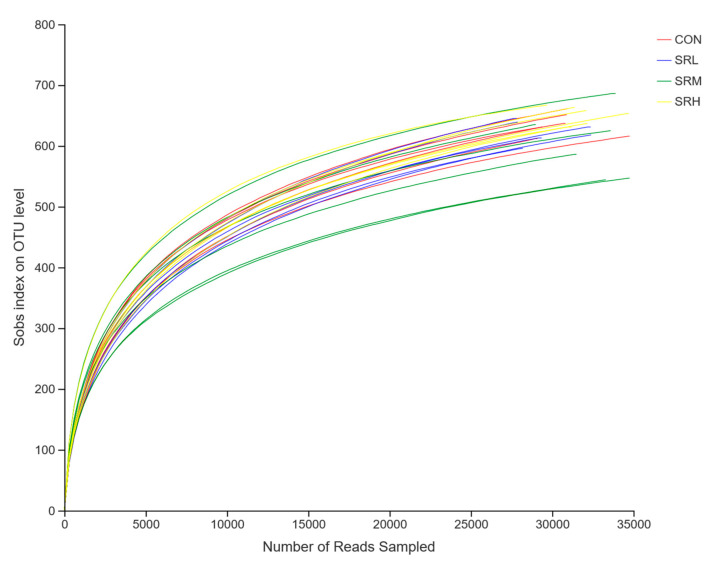
Sobs index exponential dilution curve.

**Figure 4 animals-10-02247-f004:**
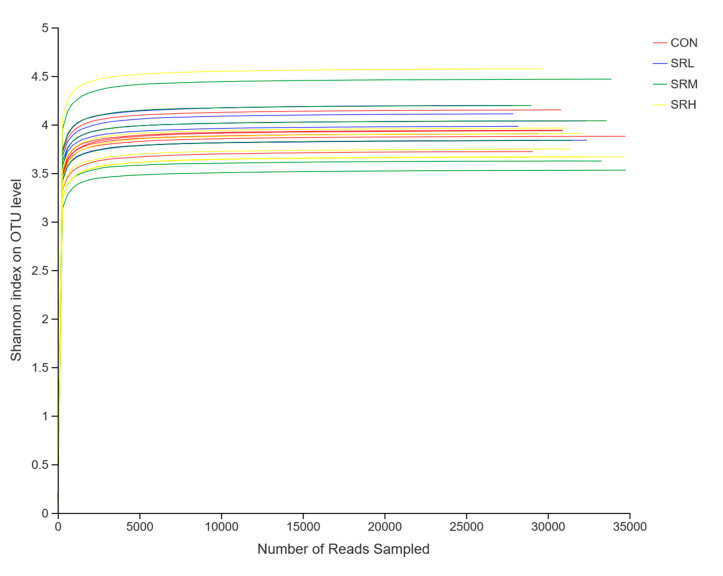
Shannon index dilution curve.

**Figure 5 animals-10-02247-f005:**
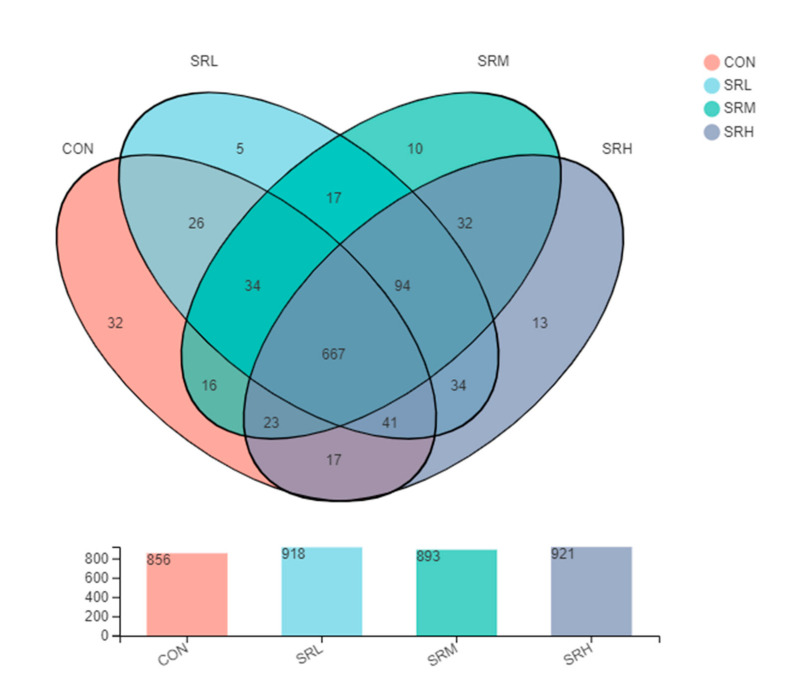
Venn diagram.

**Figure 6 animals-10-02247-f006:**
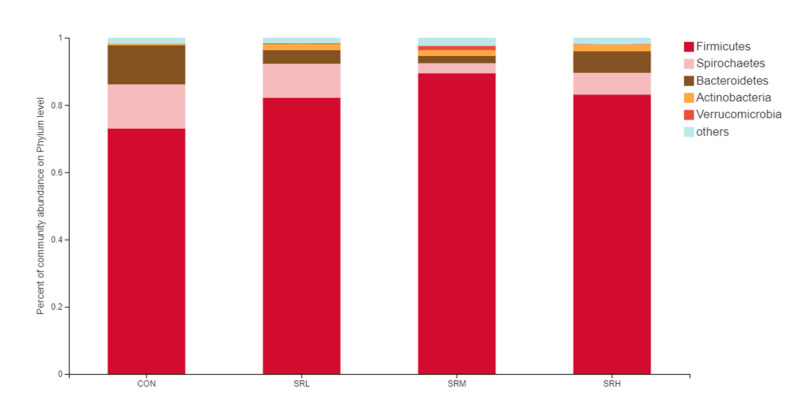
Histogram of the relative phyla abundances.

**Figure 7 animals-10-02247-f007:**
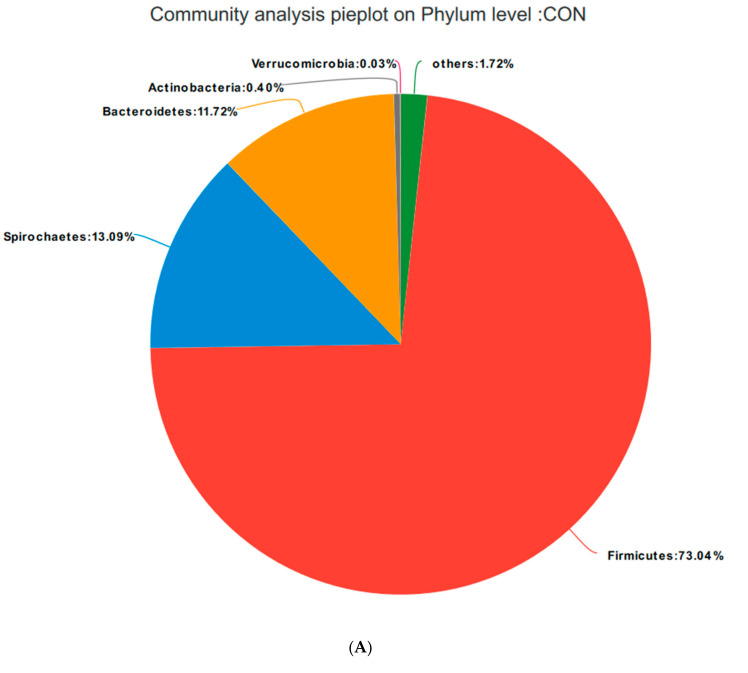
Fan chart of the relative phyla abundances. The gut bacterial composition at the phylum level in each sample. According to the results of the species notes, we selected the gut bacterial species at the phylum level in CON (**A**), SRL (**B**), SRM (**C**), and SRH (**D**). Relative abundances < 0.01% were classified as others.

**Figure 8 animals-10-02247-f008:**
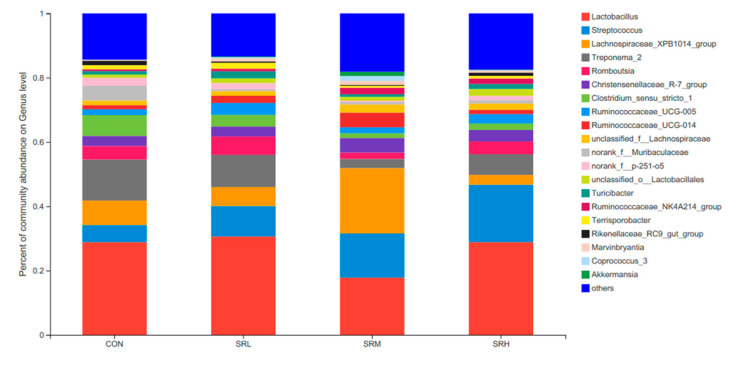
Histogram of the relative genera abundance.

**Figure 9 animals-10-02247-f009:**
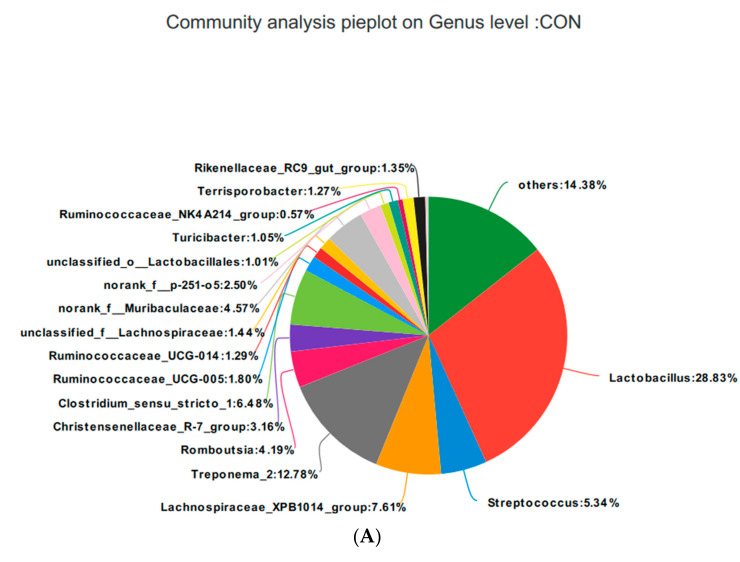
Fan chart of the relative genera abundance. The gut bacterial composition at the genus level in each sample. According to the results of the species notes, we selected the gut bacterial species at the genus level in CON (**A**), SRL (**B**), SRM (**C**), and SRH (**D**). Relative abundances < 0.01% were classified as others.

**Figure 10 animals-10-02247-f010:**
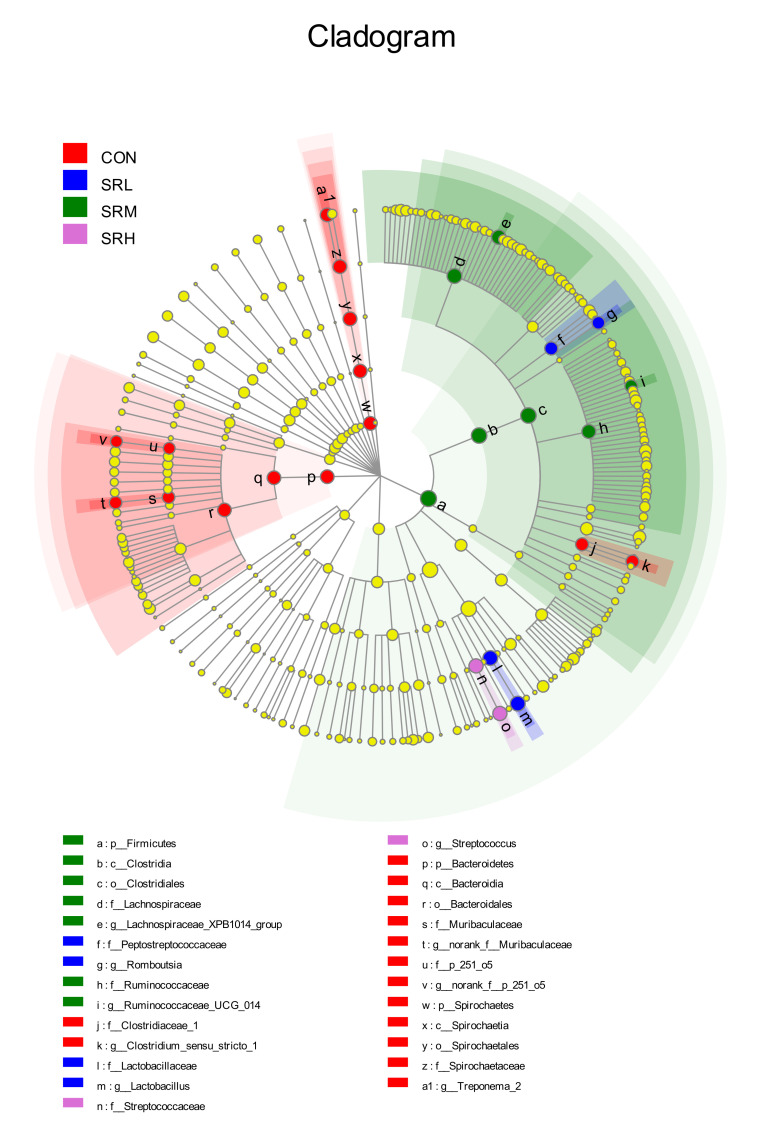
Evolution branch diagram based on the linear discriminant analysis effect size (LEfSe) analyses of the different stevia residue treatments.

**Figure 11 animals-10-02247-f011:**
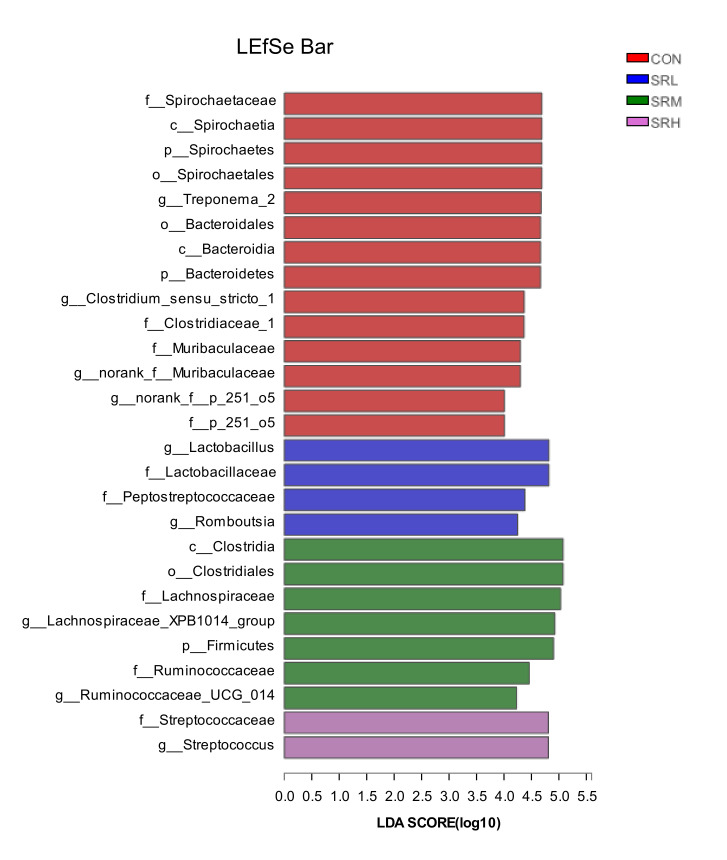
Histogram of the linear discriminant analysis (LDA) value distribution.

**Table 1 animals-10-02247-t001:** Test grouping.

Grouping	Dietary Composition Sample	No.
Control group	Regular diet	CON
Experimental diet I	20% Stevia residue	SRL
Experimental diet II	30% Stevia residue	SRM
Experimental diet III	40% Stevia residue	SRH

Note: CON: control group; SRL: low stevia residue (20%); SRM: medium stevia residue (30%); SRH: high stevia residue (40%).

**Table 2 animals-10-02247-t002:** Nutrition levels of stevia residue.

Nutrition Level	Content (%)
Dry matter (DM)	86.24
Heat energy	13.65
Crude protein (CP)	10.65
Crude fat (EE)	5.06
Crude fiber (CF)	22.97
Dietary fiber (DF)	55.10
Soluble dietary fiber (SDF)	0
Insoluble dietary fiber (IDF)	55.10
Acid detergent fiber (ADF)	66.87
Crude ash (CA)	26.05

**Table 3 animals-10-02247-t003:** Diet composition and nutrition level (air-dried basis, %).

Material	Control Group	20% Stevia Residue Group	30% Stevia Residue Group	40% Stevia Residue Group
Diet composition (%)				
Corn	61.1	36.59	40.09	39.45
Solvent rice bran meal	5	8	4	2.5
Wheat bran	10	20.2	11	1.5
Full-fat rice bran	5	8	4.5	2.5
Soybean meal	14	3	5.3	7.2
Stevia residue	0	20	30	40
Soybean oil	1.25	0.45	1.35	2.7
Stone powder	1.1	1.3	1.3	0.8
Sodium chloride	0.49	0.4	0.4	0.4
Calcium bicarbonate	0.31	0.31	0.31	1.2
Potassium chloride	0.05	0.05	0.05	0.05
Sodium bicarbonate	0.35	0.35	0.35	0.35
Choline	0.135	0.135	0.135	0.135
Lysine	0.135	0.135	0.135	0.135
Methionine	0.015	0.015	0.015	0.015
Threonine	0.05	0.05	0.05	0.05
Premix	1	1	1	1
Plant essential oil	0.015	0.015	0.015	0.015
Nutrition level				
Metabolizable energy (ME) (MJ/kg)	13.38	10.88	10.89	10.88
CP/%	14.53	11.91	11.91	11.92
CF/%	3.15	7.79	9.15	10.68
Lysine/%	0.79	0.53	0.49	0.47
Methionine/%	0.26	0.19	0.16	0.14
Methionine + cystine/%	0.53	0.38	0.33	0.29
Threonine/%	0.57	0.38	0.35	0.32
Calcium/%	0.55	0.59	0.58	0.61
Total phosphorus/%	0.56	0.62	0.43	0.46
Sodium/%	0.22	0.19	0.18	0.17
Feed cost ($)				
Feed cost/t	399.32	274.16	261.94	260.45
Feed cost/per sow d	1.20	1.10	1.05	1.04

Notes: the premix provided the following per kg of diet: VA 368,000 IU, VD3 120,000 IU, VE 2300 mg, VB1 92 mg, VB2 218 mg, pantothenic acid 920 mg, niacin 1610 mg, biotin 20.7 mg, Cu 0.74 g, Fe 8.3 g, Zn 2.8 g, and Mn 2.3 g. Lysine, methionine, and threonine additives are crystalline amino acids. Nutrition levels calculated by using the tables of feed composition and nutritive values in China (2018, twenty-ninth edition) Chinese feed database according to the chemical composition of the dietary ingredients. Crude protein and amino acids used standardized ileal digestibility to calculate. Nutrition intake per sow per day was up to the standard of Chinese ministry of agriculture of pig-breeding standards (NY/T 65-2004).

**Table 4 animals-10-02247-t004:** Experimental materials and reagents.

Reagent	Model	The Company	Countries
DNA extraction kit	E.Z.N.A.^®^ Soil DNA Kit	Omega Bio-Tek	The United States
Agarose	biowest agArose	Biowest	Spain
FastPfu Polymerase	FastPfu Polymerase	TransGen	China
AxyPrep DNA Gel Extraction Kit	Axygen Biosciences	Axygen	The United States
Library constraction kit	NEXTFLEX^®^ Rapid DNA-Seq Kit	Bioo Scientific	The United States
Sequencing kit	MiSeq Reagent Kit	Illumina	The United States

**Table 5 animals-10-02247-t005:** Experimental instruments and equipment.

The Instrument	Model	The Company	Countries
Pipettor	Eppendorf N13462C	Eppendorf	Germany
Miniature centrifuge	ABSON MiFly-6	Hefei Ebensen Scientific Instrument Co., Ltd.	China
Miniature centrifuge	Eppendorf 5430 R	Eppendorf	Germany
Highspeed table freezing centrifuge	Eppendorf 5424R	Eppendorf	Germany
Ultramicrospectrophotometer	NanoDrop2000	Thermo Fisher Scientific	The United States
ELIASA	BioTek ELx800	Biotek	The United States
Vortex mixer	QL-901	Haimen Qilin Bell Instrument Manufacturing Co., Ltd.	China
Grinding mill	TL-48R	Shanghai Wanbai Biotechnology Co., Ltd.	China
MP grinding mill	FastPrep-24 5G	MP	The United States
Microfluorometer	Quantus ™ Fluorometer	Promega	The United States
Magnetic shelf		Sangon Biological Engineering (Shanghai) Co., Ltd.	China
Electrophoresis apparatus	DYY-6C	Beijing Liuyi Instrument Factory	China
PCR amplifier	ABI GeneAmp^®^ The type 9700	ABI	The United States
Sequenator	Illumina Miseq	Illumina	The United States

**Table 6 animals-10-02247-t006:** Alpha diversity analysis.

	Group	CON	SRL	SRM	SRH
Indicators					
Sobs		634.67 ± 19.20 ^ab^	623.67 ± 18.20 ^ab^	603.83 ± 55.32 ^b^	651.50 ± 14.47 ^a^
Chao		768.76 ± 41.78 ^a^	750.80 ± 45.37 ^a^	723.27 ± 56.74 ^a^	774.28 ± 23.30 ^a^
Ace		760.31 ± 25.98 ^a^	752.19 ± 31.06 ^a^	718.69 ± 51.04 ^a^	770.22 ± 26.45 ^a^
Shannon		3.92 ± 0.14 ^a^	4.00 ± 0.15 ^a^	3.95 ± 0.36 ^a^	3.92 ± 0.34 ^a^
Simpson		0.06 ± 0.01 ^b^	0.05 ± 0.01 ^b^	0.08 ± 0.03 ^ab^	0.08 ± 0.02 ^a^
Coverage		0.99 ^a^	0.99 ^a^	0.99 ^a^	0.99 ^a^

Note: The values with the same lowercase-letter superscripts within the same row mean insignificant difference (*p* > 0.05), while with different lowercase-letter superscripts mean significant differences (*p* < 0.05).

**Table 7 animals-10-02247-t007:** Comparison of changes in genus with significant differences in each group.

Project	CON	SRL	SRM	SRH
*g__Lactobacillus*			↓	
*g__Lachnospiraceae_XPB1014_group*			↑	↓
*g__Treponema_2*			↓	↓
*g__Clostridium_sensu_stricto_1*		↓	↓	↓
*g__Christensenellaceae_R-7_group*			↑	
*g__Ruminococcaceae_UCG-005*		↑		
*g__norank_f__Muribaculaceae*		↓	↓	↓
*g__Romboutsia*			↓	
*g__Ruminococcaceae_UCG-014*		↑	↑	
*g__Rikenellaceae_RC9_gut_group*		↓	↓	
*g__unclassified_f__Lachnospiraceae*			↑	
*g__norank_f__p-251-o5*			↓	
*g__Ruminococcaceae_NK4A214_group*			↑	↑
*g__Terrisporobacter*			↓	
*g__unclassified_o__Lactobacillales*				↑

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
