# Peer review of "Effects of Dietary Supplementation with High Fiber (Stevia Residue) on the Fecal Flora of Pregnant Sows"

_animals, 2020, doi:10.3390/ani10122247_

Round 1
Reviewer 1 Report
the use of the word inhibit realtive -- inhibition can only be evaluated with a real challenge - the treatment reduced the realtive amount ( under the conditions of the experiment is understood ) --
the diets don't have the same lysine to Cal ratio -- the use of protein to energy is not equal when the essential amino acids are not in the same ratio -to energy - the diets and feeding did not result in the same amounts of essential animo acids per day and thus - the diets are less expensive per day - diets need to be balanced on SID lysine to Calorie ratio not CP to energy ratio
to compare dietary costs all nutritents ( except fiber ) should have been fed at the same amount per sow per day -
this in valdiates any cost comparisions amongst the diets - you are feed less of some more expeensive nutritents per sow per day and also caused shifts in the animo acids which may also have caused some of the changes in microbiome not just the type and amount of fiber
table 3 diet 4 - 1.2 $ cal bicarbonatre seems high realtive to the other diets why was stone pwoder redcued and ca bicarb increased in this diet almost 4 X? is this typo ?
line 350 can make better is not good English - Increased numbers of fiber-digesting bacteria can increase the ultization of dietary fiber in the large instestine of gestating sows.
Author Response
EDITOR’S COMMENTS TO AUTHORS:
-Reviewer 1
We want to begin by thanking reviewer 1. According with your advice, we amended the relevant part in manuscript. Your questions were answered below.
- the use of the word inhibit realtive -- inhibition can only be evaluated with a real challenge - the treatment reduced the realtive amount ( under the conditions of the experiment is understood ) –
Response: Thank you for the comments. But I am sorry that I cannot find the inhibit relative. In my manuscript, I used inhibit the relative abundance. I thanks for you suggestions and if you have other suggestions, please give advice or comments.
- the diets don't have the same lysine to Cal ratio -- the use of protein to energy is not equal when the essential amino acids are not in the same ratio -to energy - the diets and feeding did not result in the same amounts of essential animo acids per day and thus - the diets are less expensive per day - diets need to be balanced on SID lysine to Calorie ratio not CP to energy ratio
to compare dietary costs all nutritents ( except fiber ) should have been fed at the same amount per sow per day -
this in valdiates any cost comparisions amongst the diets - you are feed less of some more expeensive nutritents per sow per day and also caused shifts in the animo acids which may also have caused some of the changes in microbiome not just the type and amount of fiber
Response: Thank you for the comments. I will explain why the diet I designed. When we were going to design the diet, we concerned about the total intake about the nutrition level. On the basis of the control group, we tried to control the total intake of nutrients to be similar. After designing many versions of diets, we found it difficult to control the rate to be similar. Due to the low nutrition levels of stevia residue and the same contents of the diets, we had try our best to design these diets and according to the intake, the nutrition was similar.
- table 3 diet 4 - 1.2 $ cal bicarbonatre seems high realtive to the other diets why was stone pwoder redcued and ca bicarb increased in this diet almost 4 X? is this typo ?
Response: Thank you for the comments. We examined the feed formula and we were sure it was correct. Because of the lower contents of the feed, especially soybean meal, the Calcium level is so low and the other feed additive were configured and we could not change them.
- line 350 can make better is not good English - Increased numbers of fiber-digesting bacteria can increase the ultization of dietary fiber in the large instestine of gestating sows.
Response: Thank you for your reminding. We are very sorry for the unidiomatic expressions. We will change the sentences in the better way.
Lines 349-350: Increased numbers of fiber-digesting bacteria can increase the ultization of dietary fiber in the large instestine of gestating sows.

Reviewer 2 Report
The authors have complied with the majority of the comments and the quality of the manuscript has been improved.
Author Response
EDITOR’S COMMENTS TO AUTHORS:
-Reviewer
We want to thanking reviewer 2. According with your advanced advices, we amended the relevant part in manuscript and perfected the manuscript. Your previous suggestions were of great help to our future research and improving the quality of our manuscript. Also let us have a more rigorous attitude towards researches. In the future research, we will be more strict with ourselves and improve our level. Thank you again for your help in our research.
Round 2
Reviewer 1 Report
the woring in abstract should be reduce the relative abundance - which you results indicate did happen - inhibit is much stronger word and requires challenge trials that indicate actual inhibitiion not just shift in relative numbers withoiut true challenge
the diets were not formulated on equal lysine and essential animo accid basis - actually to really compare value of an ingredient one really needs to knwo th edigestiblity of the feedstoff and then the feedstuff included and the diets balanced for all key nutirients includeing SID lysine - meth - trp - thr etc whihc was not done so no statement can be made about the feed cost --- it should be discussed the the diets were not balanced for essential amino acids and that perhaps the digibiltiy of the nutrients ie AA's are not known --
feeding this high of fiber diets -- and addig another ingredient has other costs - seaprate storage bins - and increased manure handling costs and issues -- high fiber diets such as DDGG's has lead to manure foaming and barn fires -- the odors in the manure can increase and productin of methane ( and now greenhouse gas reduction policies being created) also the feeding of high fiber diets increases heat increment and heat stress in late gestation has impact on sow lactation performance - it is beyond the scope of this paper to evaluate all the secondary impacts of feeding such high fiber diets - as they diets were formulated no consideration for the esssential AA amounts - no statements can be made about feed costs --perhaps you can state in the past the addition of fiber to sows diets has increased daily feed costs - the means diets were balanced in this paper does not allow fair comparison - but stevia is a realtivily low cost feed ingredient to add additional fiber -- additional research on the digestibility of key nutrients and consideration of secondary impacts of feeding high fiber diets must be taken into acocunt - such as heat increament - increasd methane production , manure mangement etc
Author Response
EDITOR’S COMMENTS TO AUTHORS:
-Reviewer 1
We want to begin by thanking reviewer 1. According with your advice, we amended the relevant part in manuscript. Your questions were answered below.
- the woring in abstract should be reduce the relative abundance - which you results indicate did happen - inhibit is much stronger word and requires challenge trials that indicate actual inhibitiion not just shift in relative numbers withoiut true challenge
Response: Thank you for the comments. We thank you for the lack of rigor in our manuscript and felt very sorry for that. We changed the sentences and used better words.
Lines 17: reduce the relative abundance of harmful bacteria
Lines 44: reduce the relative abundance of harmful bacteria
- feeding this high of fiber diets -- and addig another ingredient has other costs - seaprate storage bins - and increased manure handling costs and issues -- high fiber diets such as DDGG's has lead to manure foaming and barn fires -- the odors in the manure can increase and productin of methane ( and now greenhouse gas reduction policies being created) also the feeding of high fiber diets increases heat increment and heat stress in late gestation has impact on sow lactation performance - it is beyond the scope of this paper to evaluate all the secondary impacts of feeding such high fiber diets - as they diets were formulated no consideration for the esssential AA amounts - no statements can be made about feed costs --perhaps you can state in the past the addition of fiber to sows diets has increased daily feed costs - the means diets were balanced in this paper does not allow fair comparison - but stevia is a realtivily low cost feed ingredient to add additional fiber -- additional research on the digestibility of key nutrients and consideration of secondary impacts of feeding high fiber diets must be taken into acocunt - such as heat increament - increasd methane production , manure mangement etc
Response: Thank you for the comments. And we will think about more for designing the diets in our future research. The nutrition levels are not same but similar, but are higher than that in the standards of NRC2012. And we will concerned about the balance for essential amino acids more in our future researches. Your advice is quite correct, and we will pay more attention to it in future research. And we concerned about other spending for high-fiber diets. But sorry for our incomplete consideration. The manner of cleaning the farm is still by people in China, so we forgot to consider labor cost of cleaning. So we thought the feed cost was reduced. Your suggestions are of great significance to our follow-up research, which will make our research more thorough and comprehensive. And we will have another manuscript for explaining the digestibility and effects of lactating sows to perfect our research. In this manuscript, we just discussed about the effects of the fecal flora.

This manuscript is a resubmission of an earlier submission. The following is a list of the peer review reports and author responses from that submission.
Round 1
Reviewer 1 Report
Major Comments
The manuscript negotiates a topic about the use of high fiber sources in sows. The manuscript has an imbalance in structure and extent in its sections. It is asked by the authors to revise extensively the discussion part in order the manuscript to be eligible for publication. A focus on the relevance of the findings in respect to the sows should be more covered. More up-to-date references regarding the use of various origin fiber sources should be used.
Minor Comments
Lines 17-18: Please revise or remove. Linking the economic benefit with reproduction and production efficiency in sows is not completely linked with the major findings outlines previously.
Line 44: “reduce the breeding cost” please remove or revise. This specific statement is not supported by the text in the abstract before.
Line 57: “reduce feed nutrients” please elaborate on this
Line 75: could the authors elaborate on the physical form and characteristics of Stevia residue
Line 89: if possible provide the name and chemical composition of the disinfection material
Line 105: this means that experimental sows were fed on a different level than the control ones. Could the authors explain why this arrangement took place? Was it due to an energy reduction of the diets?
Lines 315-317: should a reference for the effect of fiber in weaned piglets is relevant for the present study, it should mentioned in a way by the authors that such effects of fiber may have different physiological background, as the intestinal microbial community of piglets and digestive procedure are different than sows.
Lines 317-320: the same remark as previous here. The reference from Liu et al 2018, is from a study in weaned piglets. It is advised to the authors to elaborate more on studies with sows rather than piglets.
Line 323: Guo (2018) is probably missing from the references. Also specify in which animal category
Lines 346-348: these lines should be removed, as such an economic impact cannot be supported by the results of the study. Moreover to support such a conclusion more details on performance data should have been provided.
Author Response
EDITOR’S COMMENTS TO AUTHORS:
-Reviewer 1
We want to begin by thanking reviewer 1. According with your advice, we amended the relevant part in manuscript. Your questions were answered below.
- Major Comments
The manuscript negotiates a topic about the use of high fiber sources in sows. The manuscript has an imbalance in structure and extent in its sections. It is asked by the authors to revise extensively the discussion part in order the manuscript to be eligible for publication. A focus on the relevance of the findings in respect to the sows should be more covered. More up-to-date references regarding the use of various origin fiber sources should be used.
Response: Thank you for the comments. We perfected the discussion.
Line 307-358 :In recent years, the intestinal health of pigs has received greater attention. The small intestines are the site of feed digestion and nutrient absorption, undigested and indigestible substances in the small intestines are decomposed by microorganisms of large intestines. The intestines represent the largest immune organ in pigs, with 70 % of the immune cells and 60 % - 80 % of the immunoglobulin in the body. The intestinal tract is one of the largest habitats for microbes, and some have even called it a microbial organ (Yao et al., 2014). In addition to the beneficial microorganisms, harmful bacteria are also widely abundant in the intestines, accounting for 70 % of the diseases in animals (Wang, 2018; Ma et al., 2017; Zheng, 2016). Under normal conditions, the intestinal flora maintains the balance between the internal and external environments, ensures normal digestion and nutrient absorption, resists foreign pathogen invasion, and produces important metabolites and bioactive components (Caesar et al., 2010). High-fiber diets used had higher crude fiber contents, including more soluble and insoluble dietary fiber than conventional diets in pig farms. Dietary fiber is prebiotic and changes the intestinal microbiota and promotes the growth of Bifidobacteria and Lactobacillus (Davison et al., 2018). Li et al. (2019) found adding 1% Gracilaria Residue to the diet of pregnant sows can increase the number of Lactobacillus and reduce the number of Escherichia coli in rectum. Liu (2016) reported feeding pregnant Meishan sows in three different crude fiber content diets (2.5% CF group, 5% CF group and 7.5% CF group), compared with 2.5% CF group, the Operational taxonomic units (OTUs) was higher than that of 5% CF group and 7.5% CF group obviously (P<0.05), the fecal bacteria community diversity improved significantly (P=0.01), and the relative abundances of Bacillibacteria, Bacteroides, Fibrobacteres, Ruminococcus, Butyrivibrio and Lactobacillus of 7.5% CF group were significant increased than those of 2.5% CF group (P<0.05) , the relative abundances of Streptococcus and Escherichia-Shigella of 7.5% CF group were significant decreased than those of 2.5% CF group (P<0.01, P=0.04). Experiments have shown that higher fiber content can increase the abundance of Firmicutes in pregnant sows. Xu et al. (2020) found that adding 2.0% guar gum plus pregelatinized waxy maize starch (SF) in pregnant sows diets can significant increased gut bacteria community diversity and Firmicutes and Ruminococcaceae were obviously enriched in SF-fed sows (P<0.01). Zhang et al. (2017) researched that compared with the two control groups, the 1.5% inulin groups had extremely significant increase of the fecal microbial community diversity (P<0.01), the dominated phyla were significant increased including Firmicutes, Bacteroidetes, Spirochaetes, Tenericutes, and Proteobacteria (P<0.05), and had the tendency towards the increase of the relative abundance of Firmicutes/Bacteroidetes ratio (P=0.07). However, Guo (2018) found that high-fiber treatments in Tianjing black pigs can significantly improve Firmicutes abundance in the jejunum and ileum (P < 0.05) and that Firmicutes will first increase and then significantly decrease with increasing fiber levels in the cecum and duodenum (P < 0.05). It might has the same tendency for gestation sows, the reduced abundances may have been due to the exorbitant coarse fiber contents (15.5 %); the microbes in large intestines cannot digest such high concentrations of crude fiber, which may have caused microecological imbalances. The results from the above experiments are consistent with this study. Lactobacillus and Bacillus (Firmicutes) play important roles in maintaining intestinal health and promoting intestinal physiological function. Increased Firmicutes abundances suggest that high-fiber diets can maintain intestinal microecological balance. High dietary fiber levels will lead to an increase in the number of fiber-breaking bacteria (Rajesh et al., 2016). The increase in the number of fiber-breaking bacteria can make a better utilization of feed fiber in gestation sows. Wang et al. (2018) showed that castrated boars fed a 15 % alfalfa fiber diet increased the abundances of Lactobacillus and Desulfovibrio. Laitat et al. (2015) also reported that dietary supplementation with 24 % beet residue could significantly increase the relative abundance of Bifidobacteria and Lactobacillus in the intestinal tract of growing pigs while reducing Escherichia coli abundance. This is consistent with the conclusions of our study. Liu et al. (2016) found that Christensenellaceae_R-7_group abundance was higher on days 14 and 28 in weaned piglets fed brown algae glume, and Christensenellaceae_R-7_group abundance was significantly higher on day 14 (P < 0.05). This is consistent with the significant increases in g__Christensenellaceae_R-7_group abundance in this experiment. Christensenellaceae_R-7_group can regulate lipid metabolism and reduce the occurrence of obesity (Goodrich et al., 2014), so high-fiber diets may help to prevent obesity in pregnant sows.
- Lines 17-18: Please revise or remove. Linking the economic benefit with reproduction and production efficiency in sows is not completely linked with the major findings outlines previously.
Response: Thank you for the comments. We remove this sentence.
Lines 14-17: The Stevia residue supplementation reduced the feed cost, the most reduction rate was 13.33%. In conclusion, high-fiber diets can improve the intestinal balance and evenness index of the intestinal flora in pregnant sows, promote the relative abundance of beneficial bacteria, inhibit the relative abundance of harmful bacteria, reduce feed cost.
- Line 44: “reduce the breeding cost” please remove or revise. This specific statement is not supported by the text in the abstract before.
Response: We feel sorry that we did not make the correct expression of the information here. We changed this sentence.
Line 41-43: And compared with the control group, the stevia residues treatment groups reduced the feed cost 8.33%, 12.50%, and 13.33% per sow per day, respectively. In conclusion, high-fiber diets can improve the intestinal balance and evenness index of the intestinal flora in pregnant sows, promote the relative abundance of beneficial bacteria, inhibit the relative abundance of harmful bacteria, reduce feed cost.
Table 3: Table 3 Diet composition and nutrition level (air-dried basis, %)
Material |
Control group |
20% stevia residue group |
30%Stevia residue group |
40% stevia residue group |
Diet composition (%) |
||||
Corn |
61.1 |
36.59 |
40.09 |
39.45 |
Solvent rice bran meal |
5 |
8 |
4 |
2.5 |
Wheat bran |
10 |
20.2 |
11 |
1.5 |
Full-fat rice bran |
5 |
8 |
4.5 |
2.5 |
Soybean meal |
14 |
3 |
5.3 |
7.2 |
Stevia residue |
0 |
20 |
30 |
40 |
Soybean oil |
1.25 |
0.45 |
1.35 |
2.7 |
Stone powder |
1.1 |
1.3 |
1.3 |
0.8 |
Sodium chloride |
0.49 |
0.4 |
0.4 |
0.4 |
Calcium bicarbonate |
0.31 |
0.31 |
0.31 |
1.2 |
Potassium chloride |
0.05 |
0.05 |
0.05 |
0.05 |
Sodium bicarbonate |
0.35 |
0.35 |
0.35 |
0.35 |
Choline |
0.135 |
0.135 |
0.135 |
0.135 |
Lysine |
0.135 |
0.135 |
0.135 |
0.135 |
Methionine |
0.015 |
0.015 |
0.015 |
0.015 |
Threonine |
0.05 |
0.05 |
0.05 |
0.05 |
Premix |
1 |
1 |
1 |
1 |
Plant essential oil |
0.015 |
0.015 |
0.015 |
0.015 |
Nutrition level |
||||
Metabolizable energy (ME) (MJ/kg) |
13.38 |
10.88 |
10.89 |
10.88 |
CP / % |
14.53 |
11.91 |
11.91 |
11.92 |
CF / % |
3.15 |
7.79 |
9.15 |
10.68 |
Lysine / % |
0.79 |
0.53 |
0.49 |
0.47 |
Methionine / % |
0.26 |
0.19 |
0.16 |
0.14 |
Methionine + cystine /% |
0.53 |
0.38 |
0.33 |
0.29 |
Threonine / % |
0.57 |
0.38 |
0.35 |
0.32 |
Calcium / % |
0.55 |
0.59 |
0.58 |
0.61 |
Total phosphorus / % |
0.56 |
0.62 |
0.43 |
0.46 |
Sodium / % |
0.22 |
0.19 |
0.18 |
0.17 |
Feed cost ($) |
||||
Feed cost / t |
399.32 |
274.16 |
261.94 |
260.45 |
Feed cost / per sow×d |
1.20 |
1.10 |
1.05 |
1.04 |
- Line 57: “reduce feed nutrients” please elaborate on this
Response: Thank you for the comments. The meaning of reduce feed nutrients is reducing the nutrition levels in feed. Restrict pregnant sow feeding is not only reducing feed intake but reducing nutrition intake. Adding crude fiber can reduce feed nutrients.
- Line 75: could the authors elaborate on the physical form and characteristics of Stevia residue
Response: Thank you for your reminding. We are very sorry for the lack of these. We will adding these messages and the figure.
Lines 79-80: Stevia residue is a gray-black powder with a characteristic chrysanthemum odor which mixed with some straw like substance (Figure 1).
Line 108-109
Figure1 Stevia residue
- Line 89: if possible provide the name and chemical composition of the disinfection material
Response: Thanks for your suggestions. We have added this message.
Lines 94-95: pig house was cleaned every day at 07:00 and 14:00 by povidone iodine (Biocid-30, Pfizer Animal Health Products Co. LTD)
- 7. Line 105: this means that experimental sows were fed on a different level than the control ones. Could the authors explain why this arrangement took place? Was it due to an energy reduction of the diets?
Response: Thank you for your reminding. We are very sorry that this important question is unclear. The protein/energy ratios were similar between diets. The control and experimental groups were controlled to same metabolizable energy (ME) and almost balanced nutrition intake by different feed intake in the experiment.
Line 102-106:Control sows were fed 2.2 ± 0.2 kg/d from breeding until 35 d of pregnancy, 3.0 ± 0.2 kg/d from 36 d of pregnancy until delivery. The experimental groups were fed 3.0 ± 0.2 kg/d from breeding until 35 d of pregnancy, 4.0 ± 0.2 kg/d from 36 d of pregnancy until delivery. The control and experimental groups were controlled to same metabolizable energy (ME) by different feed intake
- 8. Lines 315-317: should a reference for the effect of fiber in weaned piglets is relevant for the present study, it should mentioned in a way by the authors that such effects of fiber may have different physiological background, as the intestinal microbial community of piglets and digestive procedure are different than sows.
Response: Thanks for your suggestions. We changed this reference and added another one.
Line 320-321: Li et al. (2019) found adding 1% Gracilaria Residue to the diet of pregnant sows can increase the number of Lactobacillus and reduce the number of Escherichia coli in rectum.
- Lines 317-320: the same remark as previous here. The reference from Liu et al 2018, is from a study in weaned piglets. It is advised to the authors to elaborate more on studies with sows rather than piglets.
Response: Thanks for your suggestions. We changed this reference and added another one.
Line 321-329 : Liu (2016) reported feeding pregnant Meishan sows in three different crude fiber content diets (2.5% CF group, 5% CF group and 7.5% CF group), compared with 2.5% CF group, the Operational taxonomic units (OTUs) was higher than that of 5% CF group and 7.5% CF group obviously (P<0.05), the fecal bacteria community diversity improved significantly (P=0.01), and the relative abundances of Bacillibacteria, Bacteroides, Fibrobacteres, Ruminococcus, Butyrivibrio and Lactobacillus of 7.5% CF group were significant increased than those of 2.5% CF group (P<0.05) , the relative abundances of Streptococcus and Escherichia-Shigella of 7.5% CF group were significant decreased than those of 2.5% CF group (P<0.01, P=0.04).
- Line 323: Guo (2018) is probably missing from the references. Also specify in which animal category
Response: Thanks for your suggestions. We changed the sentences and added the information of the reference.
Line 337-341 : However, Guo (2018) found that high-fiber treatments in Tianjing black pigs can significantly improve Firmicutes abundance in the jejunum and ileum (P  0.05) and that Firmicutes will first increase and then significantly decrease with increasing fiber levels in the cecum and duodenum (P  0.05). It might has the same tendency for gestation sows, the reduced abundances may have been due to the exorbitant coarse fiber contents (15.5 %);
Line - :Guo BT, 2018. Effects of high fiber diet on intestinal flora, nutrient digestibility and volatile fatty acids of Tianjing black pigs [D]. Tianjing Agricultural University.
- Lines 346-348: these lines should be removed, as such an economic impact cannot be supported by the results of the study. Moreover to support such a conclusion more details on performance data should have been provided.
Response: Thank you for your suggestion, it is very important. We added the message of feed cost and changed the sentence.
Line 362-363: Stevia residue is low cost and high yielding, which can effectively reduce the feed cost of pregnant sows.
Reviewer 2 Report
Line 9 – are you evaluating the basic science of HOW – the stevia affects the fecal microorganisms – or the relative numbers or concentration of different microflora – aren’t you measuring the end result and not HOW the stevia affected fecal microorganisms? Aren’t you evaluating the impact of Stevia on the fecal microbiome populations?
You don’t need “the results showed that “ just state Stevia residue supplementation …
Line 16 to 17 – reduce the occurrence of intestinal diseases – was this evaluated and tested in this paper? Inhibit the growth of harmful bacteria – in this trial were pigs challenged with harmful bacteria? Carefully check – from line 14 to 17 – what of these conclusions were tested and supported in this research. It seems multiple trials would need to be done to support those conclusions.
Line 17 – breeding cost reduced – the feed costs were reduced ? okay most cases if fiber is added to diet energy content of the diet decreases and more of the low energy high fiber diet must be fed and geed costs per sow per day actually increase. Were diets fed to achieve same net energy intake ? or same amount of each diet fed ? does this research look at breeding efficiency ?
Line 20 same comment as line 9 – didn’t you really evaluate relative numbers of microorganisms by family ??
Line 23 – corn soybean meal based ?
We all diets fed at the same amount or did each sow get same amount of control diet and then additional supplement ? or different diets and if different diets – the amount of each diet sows were allowed to consume and based on NE , ME of the diet ? were diets balanced for other factors energy – AA’s – Ca , P – etc ?
Line 27 and through out paper use greater instead of “higher “
Line 28 greatest (not highest)
Online 44 – does trial prove that the stevia can inhibit the growth of bad bacteria or just that in healthy animals the amount of the bad bacteria was reduced? to show inhibition it seems one must do challenge trial. Reduce breeding cost – were diets fed to achieve the same nutrient intakes (except for fiber ??) – same energy – same essential AA’s - and other nutrients?
Line 51 – most current producers increase feed intake and dietary fiber content to reduce constipation.
As far as rectal and uterine prolapses – which have been increasing – in large studies have not been associated with dietary fiber levels. ( water treatment was key factor)
Main reason for restrictive feeding is that sows will overconsume – become obese and over conditioned.
Line 56 – what is meant by high quality fiber - ? what is the standard for high quality ?
Line 57 – reduce feed nutrients—what is meant by that
Note – the nutritional requirements of lactating sows is much greater than gestating sows – these diets do not meet the demands for modern European sows for lactation. Your Chinese requirements may be for Chinese sows – but not for highly productive European sows.
Where is the lactation diet – are the diets in table 3 gestation diets -? where are the lactation diets?
Line 105 – it seems that the sows should have been fed to achieve the same key nutrient intake – energy – lysine – Ca – avial P – and the treatment diets should have been fed such at each stage of gestation.
For lactation the diets do not meet the sows lysine – essential AA and other according to NRC -European or Brazilian feeding standards.
Line 304 – to 305 – digestion and adsorption in stomach and small intestine – very little in colon – your fecal samples at the end – “don’t use general term “intestines – small or large and what segment of the small intestine ..
Line 316 – different studies took samples at different locations –
Line 321 – where – discussion is very general and needs to be specific – age of pig location of sample
Line 327 – do intestines digest fiber or do microbes in large intestine digest fiber?
Lie 346 how was best effects determined?
If one has to feed a lot more of a high fiber – low nutrient dense diet – one does not reduce costs – the animal is essentially eating the 2 to 2.5 kg of the basal diet with the added stevia – the cost per kg of diet may decrease but kgs of feed needed for same nutrient intake – so no reduction in “breeding cost “ – and what is meant by “breeding cost “ ? - cost of breeding an animal ? total feed cost for gestation ?
The paper did not measure breeding efficiency and lacks the numbers needed to evaluate breeding efficiency. The change in fecal microbiome at 56 days of gestation does not instantly change the “breeding efficiency “ of pregnant sows.
I have two general notes:
- a) The crude fiber analytical method is very robust and reproducible within and among laboratories; however, there is incomplete recovery of cellulose, hemicellulose, and lignin. Therefore, crude fiber is not considered to be an acceptable definition for dietary fiber and is not suitable for characterizing the fiber component in pig feed.
Table 2 shows composition of stevia residue but does not display soluble and insoluble dietary fiber information. The detergent procedures, although an improvement over the crude fiber method, do not recover soluble dietary fiber, such as pectins, mucilages, gums, and beta-glucans.
- b) The experimental diets need to be described in detail. Were diets supplemented with crystalline amino acids? Were diets formulated to meet the apparent or standardized ileal digestible amino acid requirements? Please include the used nutrient recommendations for diet formulation.
Line 313 – 314: I feel you are overstating your effects - the table of the experimental diets did not show any information on soluble and insoluble dietary fibers; thus, this sentence does not make sense.
Table 3: SID or AID amino acids? Please make it clear. Please replace “digestible energy” with “metabolizable energy”.
Author Response
RESPONSE TO REVIEWERS
Dear Editor:
Thank you for spending your value time on our manuscript. We appreciated the comments from you and reviewers. We have seriously read the reviews. Considering the reviewers’ positive comments and your encouragement, we have tried our best to address the criticisms of the reviewers by supplementing additional data and answering the doubts.
In the revised manuscript, descriptions of the change (red color) was made based on the reviewer's concerns.
I hope our revision has addressed all the comments from you and the reviewers. Please let me know if you have any questions or further comments.
Best regards.
Prof. Xinping Diao
EDITOR’S COMMENTS TO AUTHORS:
-Reviewer 1
We want to begin by thanking reviewer 2. According with your advice, we amended the relevant part in manuscript. Your questions were answered below.
- Line 9 – are you evaluating the basic science of HOW – the stevia affects the fecal microorganisms – or the relative numbers or concentration of different microflora – aren’t you measuring the end result and not HOW the stevia affected fecal microorganisms? Aren’t you evaluating the impact of Stevia on the fecal microbiome populations?
You don’t need “the results showed that “ just state Stevia residue supplementation …
Response: Thank you for the comments. The purpose of this experiment is investigate the effects of different supplementation levels of stevia residues in high-fiber diets on the fecal microorganisms of pregnant sows.
Lines 9-10: This study aimed to investigate the effects of different supplementation levels of stevia residues in high-fiber diets on the fecal microorganisms of pregnant sows. Stevia residue supplementation significantly increased the abundance of beneficial bacteria
- Line 16 to 17 – reduce the occurrence of intestinal diseases – was this evaluated and tested in this paper? Inhibit the growth of harmful bacteria – in this trial were pigs challenged with harmful bacteria? Carefully check – from line 14 to 17 – what of these conclusions were tested and supported in this research. It seems multiple trials would need to be done to support those conclusions.
Response: We feel sorry that we did not make the correct expression of the information here. We added more information to support this idea in Table 3 .
Lines 14-17: The Stevia residue supplementation reduced the feed cost, the most reduction rate was 13.33%. In conclusion, high-fiber diets can improve the intestinal balance and evenness index of the intestinal flora in pregnant sows, promote the relative abundance of beneficial bacteria, inhibit the relative abundance of harmful bacteria, reduce feed cost.
Table 3 : Table 3 Diet composition and nutrition level (air-dried basis, %)
Material |
Control group |
20% stevia residue group |
30%Stevia residue group |
40% stevia residue group |
Diet composition (%) |
||||
Corn |
61.1 |
36.59 |
40.09 |
39.45 |
Solvent rice bran meal |
5 |
8 |
4 |
2.5 |
Wheat bran |
10 |
20.2 |
11 |
1.5 |
Full-fat rice bran |
5 |
8 |
4.5 |
2.5 |
Soybean meal |
14 |
3 |
5.3 |
7.2 |
Stevia residue |
0 |
20 |
30 |
40 |
Soybean oil |
1.25 |
0.45 |
1.35 |
2.7 |
Stone powder |
1.1 |
1.3 |
1.3 |
0.8 |
Sodium chloride |
0.49 |
0.4 |
0.4 |
0.4 |
Calcium bicarbonate |
0.31 |
0.31 |
0.31 |
1.2 |
Potassium chloride |
0.05 |
0.05 |
0.05 |
0.05 |
Sodium bicarbonate |
0.35 |
0.35 |
0.35 |
0.35 |
Choline |
0.135 |
0.135 |
0.135 |
0.135 |
Lysine |
0.135 |
0.135 |
0.135 |
0.135 |
Methionine |
0.015 |
0.015 |
0.015 |
0.015 |
Threonine |
0.05 |
0.05 |
0.05 |
0.05 |
Premix |
1 |
1 |
1 |
1 |
Plant essential oil |
0.015 |
0.015 |
0.015 |
0.015 |
Nutrition level |
||||
Metabolizable energy (ME) (MJ/kg) |
13.38 |
10.88 |
10.89 |
10.88 |
CP / % |
14.53 |
11.91 |
11.91 |
11.92 |
CF / % |
3.15 |
7.79 |
9.15 |
10.68 |
Lysine / % |
0.79 |
0.53 |
0.49 |
0.47 |
Methionine / % |
0.26 |
0.19 |
0.16 |
0.14 |
Methionine + cystine /% |
0.53 |
0.38 |
0.33 |
0.29 |
Threonine / % |
0.57 |
0.38 |
0.35 |
0.32 |
Calcium / % |
0.55 |
0.59 |
0.58 |
0.61 |
Total phosphorus / % |
0.56 |
0.62 |
0.43 |
0.46 |
Sodium / % |
0.22 |
0.19 |
0.18 |
0.17 |
Feed cost ($) |
||||
Feed cost / t |
399.32 |
274.16 |
261.94 |
260.45 |
Feed cost / per sow×d |
1.20 |
1.10 |
1.05 |
1.04 |
- Line 17 – breeding cost reduced – the feed costs were reduced ? okay most cases if fiber is added to diet energy content of the diet decreases and more of the low energy high fiber diet must be fed and geed costs per sow per day actually increase. Were diets fed to achieve same net energy intake ? or same amount of each diet fed ? does this research look at breeding efficiency ?
Response: We feel sorry that we did not make the correct expression of the information here. We changed these wrong sentences.
Lines 14-17: The Stevia residue supplementation reduced the feed cost, the most reduction rate was 13.33%. In conclusion, high-fiber diets can improve the intestinal balance and evenness index of the intestinal flora in pregnant sows, promote the relative abundance of beneficial bacteria, inhibit the relative abundance of harmful bacteria, reduce feed cost.
- Line 20 same comment as line 9 – didn’t you really evaluate relative numbers of microorganisms by family ?
Response: Thank you for the comments. We are sorry for our imprecise expression.
Lines 18-19: This study aimed to investigate the effects of different supplementation levels of stevia residues in high-fiber diets on the fecal microorganisms of pregnant sows
- Line 23 – corn soybean meal based ?
We all diets fed at the same amount or did each sow get same amount of control diet and then additional supplement ? or different diets and if different diets – the amount of each diet sows were allowed to consume and based on NE , ME of the diet ? were diets balanced for other factors energy – AA’s – Ca , P – etc ?
Response: Thank you for your reminding. We are very sorry that this important question is unclear. The protein/energy ratios were similar between diets. The control and experimental groups were controlled to same metabolizable energy (ME) and almost balanced nutrition intake by different feed intake in the experiment. And I described this in animals and experimental design.
Line 92:And protein/energy ratios were similar between diets.
Line 102-106:Control sows were fed 2.2 ± 0.2 kg/d from breeding until 35 d of pregnancy, 3.0 ± 0.2 kg/d from 36 d of pregnancy until delivery. The experimental groups were fed 3.0 ± 0.2 kg/d from breeding until 35 d of pregnancy, 4.0 ± 0.2 kg/d from 36 d of pregnancy until delivery. The control and experimental groups were controlled to same metabolizable energy (ME) by different feed intake
- Line 27 and through out paper use greater instead of “higher “
Response: Thanks for your suggestions. We have changed the word.
Line 26: Compared to the control group, species diversity was greater in the 30 % stevia residue group.
- Line 28 greatest (not highest)
Response: Thanks for your suggestions. We have changed the word.
Line 27: At the phylum level, Firmicutes and Spirochaetes had the greatest relative abundances;
- 8. Online 44 – does trial prove that the stevia can inhibit the growth of bad bacteria or just that in healthy animals the amount of the bad bacteria was reduced? to show inhibition it seems one must do challenge trial. Reduce breeding cost – were diets fed to achieve the same nutrient intakes (except for fiber ??) – same energy – same essential AA’s - and other nutrients?
Response: Thanks for your suggestions. We are sorry that we did not list a clear description of feed cast.
We feel sorry that we did not make the correct expression of the information here. We added more information to support this idea in Table 3 and changed wrong sentence.
Line 41-45: And compared with the control group, the stevia residues treatment groups reduced the feed cost 8.33%, 12.50%, and 13.33% per sow per day, respectively. In conclusion, high-fiber diets can improve the intestinal balance and evenness index of the intestinal flora in pregnant sows, promote the relative abundance of beneficial bacteria, inhibit the relative abundance of harmful bacteria, reduce feed cost.
Table3: Table 3 Diet composition and nutrition level (air-dried basis, %)
Material |
Control group |
20% stevia residue group |
30%Stevia residue group |
40% stevia residue group |
Diet composition (%) |
||||
Corn |
61.1 |
36.59 |
40.09 |
39.45 |
Solvent rice bran meal |
5 |
8 |
4 |
2.5 |
Wheat bran |
10 |
20.2 |
11 |
1.5 |
Full-fat rice bran |
5 |
8 |
4.5 |
2.5 |
Soybean meal |
14 |
3 |
5.3 |
7.2 |
Stevia residue |
0 |
20 |
30 |
40 |
Soybean oil |
1.25 |
0.45 |
1.35 |
2.7 |
Stone powder |
1.1 |
1.3 |
1.3 |
0.8 |
Sodium chloride |
0.49 |
0.4 |
0.4 |
0.4 |
Calcium bicarbonate |
0.31 |
0.31 |
0.31 |
1.2 |
Potassium chloride |
0.05 |
0.05 |
0.05 |
0.05 |
Sodium bicarbonate |
0.35 |
0.35 |
0.35 |
0.35 |
Choline |
0.135 |
0.135 |
0.135 |
0.135 |
Lysine |
0.135 |
0.135 |
0.135 |
0.135 |
Methionine |
0.015 |
0.015 |
0.015 |
0.015 |
Threonine |
0.05 |
0.05 |
0.05 |
0.05 |
Premix |
1 |
1 |
1 |
1 |
Plant essential oil |
0.015 |
0.015 |
0.015 |
0.015 |
Nutrition level |
||||
Metabolizable energy (ME) (MJ/kg) |
13.38 |
10.88 |
10.89 |
10.88 |
CP / % |
14.53 |
11.91 |
11.91 |
11.92 |
CF / % |
3.15 |
7.79 |
9.15 |
10.68 |
Lysine / % |
0.79 |
0.53 |
0.49 |
0.47 |
Methionine / % |
0.26 |
0.19 |
0.16 |
0.14 |
Methionine + cystine /% |
0.53 |
0.38 |
0.33 |
0.29 |
Threonine / % |
0.57 |
0.38 |
0.35 |
0.32 |
Calcium / % |
0.55 |
0.59 |
0.58 |
0.61 |
Total phosphorus / % |
0.56 |
0.62 |
0.43 |
0.46 |
Sodium / % |
0.22 |
0.19 |
0.18 |
0.17 |
Feed cost ($) |
||||
Feed cost / t |
399.32 |
274.16 |
261.94 |
260.45 |
Feed cost / per sow×d |
1.20 |
1.10 |
1.05 |
1.04 |
- 9. Line 51 – most current producers increase feed intake and dietary fiber content to reduce constipation.
As far as rectal and uterine prolapses – which have been increasing – in large studies have not been associated with dietary fiber levels. ( water treatment was key factor)
Main reason for restrictive feeding is that sows will overconsume – become obese and over conditioned.
Response: Thanks for your suggestions. But I found that long-term constipation would lead to rectocele or even withdrawal of the uterus. The toxin produced by excrement fermentation could damage the organs of the body and cause a variety of inflammation: such as hysteritis, etc., but also exacerbate the sow's breast swelling phenomenon, serious would cause mastitis, mastitis and the production of toxins will cause piglets dysentery. Although constipation is the main cause rather than a lack of fibre, a high fibre diet can relieve constipation and to some extent reduce the incidence of these diseases. At the same time, more fiber can improve the water intake. And your suggestion is correct, I chose more direct diseases caused by constipation.
Lines 52-54: Constipation frequently occurs, and in severe cases, results in prolapse of the anus, systemic diseases, and stereotyped behavior, such as empty chewing, rail biting, and ground licking (Lawrence et al., 1993)
[1]Lawrence AB, Terlouw EM, 1993. A review of behavioral factors involved in the development and continued performance of stereotypic behaviors in pigs [J]. Anim Sci. 71(10):2815–25.
[2]Yu AM. Adverse effects of constipation on sows and improvement measures [J]. Animals Breeding and Feed,2020(03):71-72.
[3]Huang SP. Causes, harms and preventive measures of constipation of sows before and after delivery [J]. Livestock and Poultry Industry,2019,30(07):98.
[4] Peltoniemi O, Björkman S, Oliviero C. Parturition effects on reproductive health in the gilt and sow. Reprod Domest Anim. 2016 Oct;51 Suppl 2:36-47. doi: 10.1111/rda.12798. PMID: 27762056.
[5]Pearodwong P, Muns R, Tummaruk P. Prevalence of constipation and its influence on post-parturient disorders in tropical sows. Trop Anim Health Prod. 2016 Mar;48(3):525-31. doi: 10.1007/s11250-015-0984-3. Epub 2015 Dec 28. PMID: 26712363.
- Line 56 – what is meant by high quality fiber - ? what is the standard for high quality ?
Response: Thanks for your suggestions. We have detected the bacterial load in the intestine. The following modifications were added to the manuscript.
Line 56-60: Stevia residue is the waste of Stevia after stevioside extraction, dehydration and high-temperature drying. Most stevia residue is landfilled and burned, and a small part is made into fertilizer, which caused great waste and environment pollution (Zhang et al., 2007). Stevia residue has low price, high-fiber content, high-crude-protein content and long shelf-life, is a source of high-quality green crude fiber, and crude fiber can accelerate intestinal peristalsis and emptying (Zhu et al., 2010)
- Line 57 – reduce feed nutrients—what is meant by that
Note – the nutritional requirements of lactating sows is much greater than gestating sows – these diets do not meet the demands for modern European sows for lactation. Your Chinese requirements may be for Chinese sows – but not for highly productive European sows.
Where is the lactation diet – are the diets in table 3 gestation diets -? where are the lactation diets?
Response: Thanks for your suggestions. Your opinion is very helpful to us, and we very much agree with your opinion. And our sows were Danish sows, they had highly productive. The diets in table 3 are the gestation, we used them only in pregnancy. We would change a highly nutrition diet for lactation diet, which the four groups used same. And in the paper, I only described the effects of different supplementation levels of stevia residues in high-fiber diets on the fecal microorganisms of pregnant sows. Therefore I did not describe the lactation diet. The meaning of reduce feed nutrients is reducing the nutrition levels in feed. Restrict pregnant sow feeding is not only reducing feed intake but reducing nutrition intake. Adding crude fiber can reduce feed nutrients.
- Line 105 – it seems that the sows should have been fed to achieve the same key nutrient intake – energy – lysine – Ca – avial P – and the treatment diets should have been fed such at each stage of gestation.
For lactation the diets do not meet the sows lysine – essential AA and other according to NRC -European or Brazilian feeding standards.
Response: Thank you for your suggestion, it is very important. Due to your suggestion, I found the shortcomings in my current work. I will follow your suggestions to describe clearly about these. Thank you for your reminding. We are very sorry that this important question is unclear. The protein/energy ratios were similar between diets. The control and experimental groups were controlled to same metabolizable energy (ME) and almost balanced nutrition intake by different feed intake in the experiment.
Line 102-106:Control sows were fed 2.2 ± 0.2 kg/d from breeding until 35 d of pregnancy, 3.0 ± 0.2 kg/d from 36 d of pregnancy until delivery. The experimental groups were fed 3.0 ± 0.2 kg/d from breeding until 35 d of pregnancy, 4.0 ± 0.2 kg/d from 36 d of pregnancy until delivery. The control and experimental groups were controlled to same metabolizable energy (ME) by different feed intake
- Line 304 – to 305 – digestion and adsorption in stomach and small intestine – very little in colon – your fecal samples at the end – “don’t use general term “intestines – small or large and what segment of the small intestine ..
Response: Thank you for your suggestion, it is very important. Due to your suggestion, I found the shortcomings in my paper. And we described this information clearly.
Line 309-312:The small intestines are the site of feed digestion and nutrient absorption, undigested and indigestible substances in the small intestines are decomposed by microorganisms of large intestines. The intestines represent the largest immune organ in pigs,
- Line 316 – different studies took samples at different locations –
Response: Thanks for your suggestions. We changed this reference and added another one.
Line 322-323: Li et al. (2019) found adding 1% Gracilaria Residue to the diet of pregnant sows can increase the number of Lactobacillus and reduce the number of Escherichia coli in rectum.
- Line 321 – where – discussion is very general and needs to be specific – age of pig location of sample
Response: Thank you for your suggestion, it is very important. Due to your suggestion, I found the shortcomings in my paper. And we described this information clearly. We changed the discussion.
Line 331-343: Experiments have shown that higher fiber content can increase the abundance of Firmicutes in pregnant sows. Xu et al. (2020) found that adding 2.0% guar gum plus pregelatinized waxy maize starch (SF) in pregnant sows diets can significant increased gut bacteria community diversity and Firmicutes and Ruminococcaceae were obviously enriched in SF-fed sows (P0.01). Zhang et al. (2017) researched that compared with the two control groups, the 1.5% inulin groups had extremely significant increase of the fecal microbial community diversity (P0.01), the dominated phyla were significant increased including Firmicutes, Bacteroidetes, Spirochaetes, Tenericutes, and Proteobacteria (P0.05), and had the tendency towards the increase of the relative abundance of Firmicutes/Bacteroidetes ratio (P=0.07). However, Guo (2018) found that high-fiber treatments in Tianjing black pigs can significantly improve Firmicutes abundance in the jejunum and ileum (P  0.05) and that Firmicutes will first increase and then significantly decrease with increasing fiber levels in the cecum and duodenum (P  0.05). It might has the same tendency for gestation sows, the reduced abundances may have been due to the exorbitant coarse fiber contents (15.5 %);
- Line 327 – do intestines digest fiber or do microbes in large intestine digest fiber?
Response: Thank you for your suggestion, it is very important. Due to your suggestion, I found the shortcomings in my paper. And we described this information clearly.
Line 344: the microbes in large intestines cannot digest such high concentrations of crude fiber.
- Line 346 how was best effects determined?
If one has to feed a lot more of a high fiber – low nutrient dense diet – one does not reduce costs – the animal is essentially eating the 2 to 2.5 kg of the basal diet with the added stevia – the cost per kg of diet may decrease but kgs of feed needed for same nutrient intake – so no reduction in “breeding cost “ – and what is meant by “breeding cost “ ? - cost of breeding an animal ? total feed cost for gestation ?
The paper did not measure breeding efficiency and lacks the numbers needed to evaluate breeding efficiency. The change in fecal microbiome at 56 days of gestation does not instantly change the “breeding efficiency “ of pregnant sows.
I have two general notes:
- a) The crude fiber analytical method is very robust and reproducible within and among laboratories; however, there is incomplete recovery of cellulose, hemicellulose, and lignin. Therefore, crude fiber is not considered to be an acceptable definition for dietary fiber and is not suitable for characterizing the fiber component in pig feed.
Table 2 shows composition of stevia residue but does not display soluble and insoluble dietary fiber information. The detergent procedures, although an improvement over the crude fiber method, do not recover soluble dietary fiber, such as pectins, mucilages, gums, and beta-glucans.
- b) The experimental diets need to be described in detail. Were diets supplemented with crystalline amino acids? Were diets formulated to meet the apparent or standardized ileal digestible amino acid requirements? Please include the used nutrient recommendations for diet formulation.
Response: Thank you for your suggestion, it is very important. Due to your suggestion, I found the shortcomings in my paper. And we described this information clearly. First I want to explain the best effects. Considering the changes of fecal flora among groups, the 30% Stevia residue supplement group had better diversity and uniformity of fecal flora. Compared with the control group, the relative abundance of beneficial bacteria increased, the harmful bacteria decreased, and the feed cost was low. Therefore, I thought the best group was the 30% Stevia residue supplement group. Second, the information of feed cost was added in the Table 3. The meaning of breeding cost was feed cost. Stevia residue supplementation could reduce the feed cost significantly. Third, I added the information of Stevia residue in Table 2. But the time is not enough to finish the new test. And I can submit the new data on Nov. 5. I must submit my new paper before the time. I am so ashamed for wasting more time. And I will add the data later. However, we did not describe the effects of SDF and IDF on the fecal flora of pregnant sows respectively. We did not add the nutrition levels of SDF and IDF in diets. Forth, I added the information of experimental diets. The premix was mixed by Heilongjiang Anyou Biotechnology Co., Ltd. Its formulation is the company secret. The only information we got was nutrition levels. Diets formulation were up to the standard of Chinese ministry of agriculture of pig breeding standards (NY/T 65-2004).
Table 3
Table 3 Diet composition and nutrition level (air-dried basis, %)
Material |
Control group |
20% stevia residue group |
30%Stevia residue group |
40% stevia residue group |
Diet composition (%) |
||||
Corn |
61.1 |
36.59 |
40.09 |
39.45 |
Solvent rice bran meal |
5 |
8 |
4 |
2.5 |
Wheat bran |
10 |
20.2 |
11 |
1.5 |
Full-fat rice bran |
5 |
8 |
4.5 |
2.5 |
Soybean meal |
14 |
3 |
5.3 |
7.2 |
Stevia residue |
0 |
20 |
30 |
40 |
Soybean oil |
1.25 |
0.45 |
1.35 |
2.7 |
Stone powder |
1.1 |
1.3 |
1.3 |
0.8 |
Sodium chloride |
0.49 |
0.4 |
0.4 |
0.4 |
Calcium bicarbonate |
0.31 |
0.31 |
0.31 |
1.2 |
Potassium chloride |
0.05 |
0.05 |
0.05 |
0.05 |
Sodium bicarbonate |
0.35 |
0.35 |
0.35 |
0.35 |
Choline |
0.135 |
0.135 |
0.135 |
0.135 |
Lysine |
0.135 |
0.135 |
0.135 |
0.135 |
Methionine |
0.015 |
0.015 |
0.015 |
0.015 |
Threonine |
0.05 |
0.05 |
0.05 |
0.05 |
Premix |
1 |
1 |
1 |
1 |
Plant essential oil |
0.015 |
0.015 |
0.015 |
0.015 |
Nutrition level |
||||
Metabolizable energy (ME) (MJ/kg) |
13.38 |
10.88 |
10.89 |
10.88 |
CP / % |
14.53 |
11.91 |
11.91 |
11.92 |
CF / % |
3.15 |
7.79 |
9.15 |
10.68 |
Lysine / % |
0.79 |
0.53 |
0.49 |
0.47 |
Methionine / % |
0.26 |
0.19 |
0.16 |
0.14 |
Methionine + cystine /% |
0.53 |
0.38 |
0.33 |
0.29 |
Threonine / % |
0.57 |
0.38 |
0.35 |
0.32 |
Calcium / % |
0.55 |
0.59 |
0.58 |
0.61 |
Total phosphorus / % |
0.56 |
0.62 |
0.43 |
0.46 |
Sodium / % |
0.22 |
0.19 |
0.18 |
0.17 |
Feed cost ($) |
||||
Feed cost / t |
399.32 |
274.16 |
261.94 |
260.45 |
Feed cost / per sow×d |
1.20 |
1.10 |
1.05 |
1.04 |
Note: a the premix provided the following per kg of diet: VA 368,000 IU,VD3 120,000 IU, VE 2300mg, VB1 92 mg,VB2 218mg, Pantothenic acid 920 mg, niacin 1610 mg, Biotin 20.7mg, Cu 0.74 g, Fe 8.3 g, Zn 2.8g,Mn 2.3g. Lysine, Methionine, and Threonine additives are crystalline amino acids.
b Nutrition levels calculated using the tables of feed composition and nutritive values in China (2018 twenty-ninth edition) Chinese feed database according to the chemical composition of the dietary ingredients. Crude protein and amino acids used standardized ileal digestibility to calculate.
c Nutrition intake per sow per day was up to the standard of Chinese ministry of agriculture of pig breeding standards (NY/T 65-2004).
- Line 313 – 314: I feel you are overstating your effects - the table of the experimental diets did not show any information on soluble and insoluble dietary fibers; thus, this sentence does not make sense.
Response: Thank you for your suggestion. I will follow your suggestions to describe clearly about these. Thank you for your reminding. In this sentence, I did not want to describe my experimental diets. I described all kinds of high-fiber diets have more fiber than normal diets.
Line 319-320 : High-fiber diets had higher crude fiber contents, including more soluble and insoluble dietary fiber than conventional diets in pig farms.
- Table 3: SID or AID amino acids? Please make it clear. Please replace “digestible energy” with “metabolizable energy”.
Response: Thank you for your suggestion. We added the data needed. And we changed the information in Table 3.
Table 3: Table 3 Diet composition and nutrition level (air-dried basis, %)
Material |
Control group |
20% stevia residue group |
30%Stevia residue group |
40% stevia residue group |
Diet composition (%) |
||||
Corn |
61.1 |
36.59 |
40.09 |
39.45 |
Solvent rice bran meal |
5 |
8 |
4 |
2.5 |
Wheat bran |
10 |
20.2 |
11 |
1.5 |
Full-fat rice bran |
5 |
8 |
4.5 |
2.5 |
Soybean meal |
14 |
3 |
5.3 |
7.2 |
Stevia residue |
0 |
20 |
30 |
40 |
Soybean oil |
1.25 |
0.45 |
1.35 |
2.7 |
Stone powder |
1.1 |
1.3 |
1.3 |
0.8 |
Sodium chloride |
0.49 |
0.4 |
0.4 |
0.4 |
Calcium bicarbonate |
0.31 |
0.31 |
0.31 |
1.2 |
Potassium chloride |
0.05 |
0.05 |
0.05 |
0.05 |
Sodium bicarbonate |
0.35 |
0.35 |
0.35 |
0.35 |
Choline |
0.135 |
0.135 |
0.135 |
0.135 |
Lysine |
0.135 |
0.135 |
0.135 |
0.135 |
Methionine |
0.015 |
0.015 |
0.015 |
0.015 |
Threonine |
0.05 |
0.05 |
0.05 |
0.05 |
Premix |
1 |
1 |
1 |
1 |
Plant essential oil |
0.015 |
0.015 |
0.015 |
0.015 |
Nutrition level |
||||
Metabolizable energy (ME) (MJ/kg) |
13.38 |
10.88 |
10.89 |
10.88 |
CP / % |
14.53 |
11.91 |
11.91 |
11.92 |
CF / % |
3.15 |
7.79 |
9.15 |
10.68 |
Lysine / % |
0.79 |
0.53 |
0.49 |
0.47 |
Methionine / % |
0.26 |
0.19 |
0.16 |
0.14 |
Methionine + cystine /% |
0.53 |
0.38 |
0.33 |
0.29 |
Threonine / % |
0.57 |
0.38 |
0.35 |
0.32 |
Calcium / % |
0.55 |
0.59 |
0.58 |
0.61 |
Total phosphorus / % |
0.56 |
0.62 |
0.43 |
0.46 |
Sodium / % |
0.22 |
0.19 |
0.18 |
0.17 |
Feed cost ($) |
||||
Feed cost / t |
399.32 |
274.16 |
261.94 |
260.45 |
Feed cost / per sow×d |
1.20 |
1.10 |
1.05 |
1.04 |
Note: a the premix provided the following per kg of diet: VA 368,000 IU,VD3 120,000 IU, VE 2300mg, VB1 92 mg,VB2 218mg, Pantothenic acid 920 mg, niacin 1610 mg, Biotin 20.7mg, Cu 0.74 g, Fe 8.3 g, Zn 2.8g,Mn 2.3g. Lysine, Methionine, and Threonine additives are crystalline amino acids.
b Nutrition levels calculated using the tables of feed composition and nutritive values in China (2018 twenty-ninth edition). Chinese feed database according to the chemical composition of the dietary ingredients. Crude protein and amino acids used standardized ileal digestibility to calculate.
c Nutrition intake per sow per day was up to the standard of Chinese ministry of agriculture of pig breeding standards (NY/T 65-2004).
